

# Development and evaluation of pollen source methodologies for the Victorian Grass Pollen Emissions Module VGPEM1.0

Kathryn M. Emmerson[1], Jeremy D. Silver[2], Edward Newbigin[3], Edwin R. Lampugnani[3], Cenk Suphioglu[4], Alan Wain[5], and Elizabeth Ebert[5]

[1]Climate Science Centre, CSIRO Oceans & Atmosphere, Aspendale. VIC 3195, Australia
[2]School of Earth Sciences, University of Melbourne, VIC 3010, Australia.
[3]School of BioSciences, University of Melbourne, VIC 3010, Australia.
[4]School of Life and Environmental Sciences, Deakin University, Waurn Ponds, VIC 3216, Australia
[5]Bureau of Meteorology, Docklands, VIC 3008, Australia

**Correspondence:** K.M. Emmerson (kathryn.emmerson@csiro.au)

**Abstract.** We present the first representation of grass pollen in a 3D dispersion model anywhere in Australia, tested using observations from eight counting sites in Victoria. The region's population has high rates of allergic rhinitis and asthma, and this has been linked to the high incidence of grass pollen allergy. Despite this, grass pollen dispersion in the Australian atmosphere has not been studied previously, and its source strength is untested. We describe ten pollen emission source methodologies examining the strengths of different immediate and seasonal timing functions, and spatial distribution of the sources. The timing function assumes a smooth seasonal term, modulated by an hourly meteorological function. A simple Gaussian representation of the pollen season worked well (average r=0.54), but lacks the spatial and temporal variation that the satellite-derived Enhanced Vegetation Index (EVI) data can provide. However poor results were obtained using the EVI gradient (average r=0.35), which gives the timing when grass turns from maximum greenness to a drying and flowering period; this is due to the greater spatial and temporal variability from this combined spatial and seasonal term. Better results were obtained using statistical methods that combine elements of the EVI dataset, a smooth seasonal term and instantaneous variation based on historical grass pollen observations (average r=0.69). The seasonal magnitude is inferred from the maximum winter-time EVI, while the timing of the peak of the season was based on the day of the year when the EVI falls to 0.05 below its winter maximum. Measurements are vital to monitor changes in the pollen season, and the new pollen measurement sites in the Victorian network should be maintained.

*Copyright statement.* TEXT

# 1 Introduction

Pollen is a biological particle, produced by plants to transfer haploid genetic material during reproduction. With allergenic properties pollen can be a human irritant, and is strongly linked to seasonal allergic rhinitis and asthma. Whilst allergic effects




are experienced worldwide, the highest concentrations of allergic rhinitis sufferers live in Melbourne, Victoria, on the south east coast of Australia (Bousquet et al., 2008). Melbourne is a city of approximately 4.9 million inhabitants (ABS, 2018). Prior to 2017 pollen forecasts in Melbourne were generated manually by aeroallergen scientists, relying on persisting the previous day's pollen count and an interpretation of forecasted weather (Schäppi et al., 1998). Yet despite the availability of a simple

pollen forecast, they were not connected to the likelihood of thunderstorms, which proved fatal on the afternoon of November 21st 2016 (Lindstrom et al., 2017). During rush hour, a north-south line of thunderstorms developed west of Melbourne, and swept eastwards across the city. In the following hours, 9900 people visited hospitals with breathing difficulties overwhelming the emergency services. It is possible that strong winds collected large quantities of grass pollen from north western pasture regions, which were concentrated along the edge of the gust front. Victoria had experienced the world's largest epidemic

thunderstorm asthma event. In the aftermath, the State government funded better planning of health-care resources to improve preparedness and response arrangements for similar events in future (Davies et al., 2017; Lindstrom et al., 2017). This plan included development of a pilot thunderstorm asthma early warning service using statistical pollen forecasts (Silver et al., 2019), operated by the Bureau of Meteorology (BoM) and concurrent development of a pollen forecasting system, built around pollen emission and transport modelling. The pollen emissions component is called the Victorian Grass Pollen Emissions

Module version 1.0 (VGPEM1.0).

Pollen is generally not included in air quality models because its atmospheric lifetime is usually too short to be of interest. Recently human exposure to pollen has become a focus, particularly in the northern hemisphere (e.g. Sofiev et al., 2015; Zhang et al., 2014) and urban areas (Skjøth et al., 2013), such that detailed vegetation taxa maps are being produced for pollen forecasting (McInnes et al., 2017).

Techniques to model atmospheric concentrations of pollen have included statistical techniques and dispersion models. Statistical techniques using standard multiple regression analyses have predicted whether airborne pollen concentrations will be higher or lower than a long term mean with > 87 % accuracy (Smith and Emberlin, 2006), but require decade-long datasets (Emberlin et al., 2007), and are improved by the availability of multiple sampling sites. Statistical models require a representation of the flowering season, but perform poorly if the timing of the flowering season changes in different seasons (Beggs

et al., 2015), or in urban areas subject to local scale turbulence and heat island effects (Emberlin and Norrishill, 1991). Meteorological dispersion models can capture these effects, but are more computationally demanding to run. Physical dispersion of pollen includes 1) emission from the pollen source regions, 2) atmospheric transport and 3) deposition, using measurements to validate the predictions.

Kawashima and Takahashi (1999) were amongst the first to develop a numerical description of pollen within a dispersion

model, using a flowering map to simulate the cedar pollen season in the Tohoku district of Japan. In the USA, the Biogenic Emission Inventory System was adapted to emit birch and ragweed pollen and predicted the timing of the birch pollen peak to within two days (Efstathiou et al., 2011). Wozniak and Steiner (2017) modelled pollen from 13 different taxa based on plant functional type mapping for the USA, which could be used on climatic timescales. In Europe, Helbig et al. (2004) simulated hazel and alder pollen emissions and transport across Germany, but did not verify their predictions as no pollen measurements

were available. Schueler and Schlunzen (2006) simulated oak pollen emission in northern Germany by incorporation of land-



scape structural mapping, finding oak pollen plumes transported up to 100 km away. The EMPOL1.0 model for birch pollen across all of Europe was comprehensively evaluated by Zink et al. (2013). The Finnish Meteorological Institute has developed the System for Integrated modeLling of Atmospheric coMposition (SILAM; Sofiev, 2017, and references therein), to calculate concentrations of six pollen species at 10 km resolution on an hourly basis for all of Europe. This group have found the most

important input parameter is temperature (Siljamo et al., 2013).

Studies of pollen have focused on those taxa having a high allergenic burden which differs depending on region. In Europe, birch tree pollen is the major allergen and has been the focus of intense research activity (Siljamo et al., 2013; Sofiev et al., 2013; Siljamo et al., 2007; Sofiev et al., 2006). However birch is not common to Australia. Ragweed pollen, which is a common allergic trigger in the northern hemisphere, grows in the northeast and east of Australia but not elsewhere (Bass et al.,

2000). The native Australian vegetation produces little pollen (Smart et al., 1979), and there are no large forests of introduced allergenic tree species. European settlers introduced ryegrass (*Lolium perenne*) to southern Australia as they thought it superior for agricultural pasture. Ryegrass is wind-pollinated (i.e. not self- or vector-pollinated) and produces large volumes of pollen in spring. In Southern Australia most of the allergenic burden has been attributed to these pasture grasses via skin prick tests (Girgis et al., 2000; Bellomo et al., 1992). Further, the rupturing of ryegrass pollen grains releases much smaller starch particles

capable of causing asthma (Taylor and Jonsson, 2004; Suphioglu et al., 1992). VGPEM1.0 therefore focuses on pasture grass and has the following goals. The first is to improve public health emergency planning and response arrangements around thunderstorm asthma, by providing a tool for appropriate information providers (i.e. Melbourne Pollen Count and Victorian Department of Health and Human Services). VGPEM will also feed into other forecasting models like BoM's thunderstorm asthma forecast.

This paper documents the first representation of grass pollen in a 3D dispersion model anywhere in Australia. As the greatest uncertainty is the pollen emission characteristics, we develop and evaluate 10 methodologies, using observations from eight counting sites in Victoria. First we describe these grass pollen observations, and determine their correlations with observed meteorological variables. Second, the grass pollen emission methodologies are described and tested at a spatial resolution of 3 km. The best performing method is recommended for VGPEM1.0.

**2    Observations and characteristics of grass pollen**

Despite Australians having high rates of asthma and allergy compared to other Western nations (Lai et al., 2009), there were few Australian pollen observation sites for routine monitoring or research in 2016 (Beggs et al., 2015). In Australia, all pollen sampling is performed using Burkard volumetric pollen traps (de Morton et al., 2011). Samples are histologically stained and counted manually under a microscope by trained personnel who reference the samples to pollen taxonomic standards. One

limitation of this method is that pollen cannot be classified into particular species, or even genus, based on visual examination alone.

The University of Melbourne (UoM) operated a pollen count site in Victoria sporadically from the late 1970s to 1990, but since 1991 has counted annually over the three-month period of October, November and December, coinciding with the grass



pollen season (Ong et al., 1995). Whilst about 70 % of the total pollen measured at the UoM site is *Cupressaceae* from a nearby cemetery (Haberle et al., 2014), we concentrate on the *Poaceaea* (grass) pollen, as it is the dominant outdoor human allergen in Australia. The amount of grass pollen season in any particular year in Melbourne is strongly related to the amount of spring rainfall, which promotes grass growth and flowering (de Morton et al., 2011). The cumulative grass pollen count over the

Oct-Dec season in Melbourne ranges between 1500-5000 grains m⁻³, with daily maximums reaching 400 grains m⁻³ (Medek et al., 2016). In Melbourne, the highest pollen counts are usually associated with northerly continental air masses (de Morton et al., 2011), with an evening peak coinciding with the onset of the stable nocturnal boundary layer and descending air (Ong et al., 1995).

Two other pollen counting sites close to Melbourne at Burwood and Geelong have been in operation since 2012 by Deakin
University. Five new sites were introduced in 2017 around Victoria situated within university or hospital grounds (Figure 1a,b and Table 1). Pollen sampling occurred daily during the 2017 grass pollen season at 9am, representing the mean daily pollen concentration from 09:00h the previous day to 08:59h on the day of collection. Pollen observations from these eight sites are used to assess the accuracy of pollen predictions in this study.

In Australia, grass pollen counts are graded 'low' if the count is less than 20 m⁻³, 'moderate' if between 20-49 m⁻³, 'high'
if between 50-99 m⁻³ and 'extreme' if above 100 m⁻³. Between 20 and 60 days in each Melbourne season are observed in the moderate or above category, and up to 37 days in the high or above category (Medek et al., 2016). However it is clear climate change is impacting the timing and strength of the grass pollen season (Ziska and Beggs, 2012), as are changes to agricultural practices and the expanding boundary of the city. These changes highlight the importance of long term observations and the need to sustain the new pollen observation sites in Victoria.

# 3   Treatment of pollen in VGPEM1.0

Pollen is set-up in VGPEM1.0 as an inert particle tracer. The pollen source methodologies are tested using the CSIRO Chemical Transport Model (C-CTM), a framework of modules designed to calculate concentrations of gases and aerosol which are subjected to emission, dispersion and deposition within the atmosphere (Cope et al., 2009). The C-CTM has been used to model impacts of anthropogenic emissions on urban air sheds (Chambers et al., 2019; Paton-Walsh et al., 2018), volatile
organic compounds from vegetation (Emmerson et al., 2018, 2016) and also used to investigate health impacts by reducing the sulfur content in shipping fuels (Broome et al., 2016). The C-CTM is driven by meteorology from the Australian Community Climate and Earth System Simulator model (ACCESS, Puri et al., 2013), run at 3 km resolution using boundary conditions from ERA-Interim for a domain covering Victoria (Figure 1a). ACCESS provides the meteorological parameters necessary for pollen emission and transport, namely wind speed and direction, temperature, relative humidity (RH) and rainfall.

Particles are output as $\mu$g m⁻³ in the C-CTM, and require unit conversion to calculate grains m⁻³ (consistent with the pollen observations), using the mass of 1 pollen grain. Grass pollen diameters are found in the range $30 - 40$ $\mu$m (Brown and Irving, 1973). Early calculations by Smart et al. (1979) estimated the mass of one ryegrass pollen grain in Melbourne to be $1 \times 10^{-9}$ g, which converts to a very low density of 44.5 kg m⁻³ using a 35 $\mu$m diameter. The grass pollen density is a large source of



uncertainty. Whilst Smart's study is local to our work, studies of pollen from other grass taxa yield much higher densities, for example 980 kg m$^{-3}$ for *Secale* (rye) (Durham, 1946) and *Dactylis glomerata* (Stanley and Linskens, 1974).

The pollen density also impacts on the dry deposition velocity, which controls the length of time the pollen grain is airborne. The C-CTM dry deposition parameter follows Stoke's law. Sugita et al. (1999) measured *Gramineae* (grass) pollen with a fall

speed of 3.5 cm s$^{-1}$. Skjøth et al. (2007) suggest the deposition of grass pollen is four times larger than the 1 cm s$^{-1}$ estimated for birch pollen, and consistent with the 4.3 cm s$^{-1}$ measured by Durham (1946) on *Secale* (rye). We will assume each pollen particle is 35 $\mu$m in diameter, is spherical and has a density of 1000 kg m$^{-3}$, consistent with values used by Melbourne based researchers (de Morton et al., 2011; Knox, 1993), and similar to the grass pollen density used in Zhang et al. (2014). A 35 $\mu$m particle with a density of 1000 kg m$^{-3}$ yields a deposition velocity of 4.6 cm s$^{-1}$ which is similar to Skjøth et al. (2013). Using

these values, the estimated mass of each pollen grain is $22.4 \times 10^{-9}$ g.

This work relates exclusively to forecasting the presence of intact grass pollen grains in the air, within Victoria, Australia and does not consider thunderstorm cells nor the interactions of grass pollen grains within them. The process of re-entrainment of pollen grains once they are deposited to the ground is not considered, nor is the rupturing process that releases the allergenic contents of the grains - present on small starch particles. Whilst the impacts of pollen rupturing on numbers of cloud conden-

sation nuclei has been investigated by Wozniak et al. (2018), ruptured pollen grains are not routinely monitored in Victoria. Future development of VGPEM may incorporate some of these processes.

### 3.1   Pollen emissions framework

Pollen emission and transport has never been modelled in Australia, therefore we trial three different emission frameworks and vary their inputs. In some instances we test parameters proven not to work elsewhere and for other pollen taxa, to investigate

whether Australian ryegrass pollen characteristics are different. The first framework is a spatio-temporal decomposition of factors, the second is a pollen production-loss model and the third is a derivative of the statistical model for daily grass pollen concentrations used in the BoM's pilot forecasting system (Silver et al., 2019). The pollen emission rate $E$ at grid-point $(x, y)$ and time $t$ is expressed as:

$$E(x, y, t) = I(x, y, t) \times G(x, y, t) \times S(x, y) \tag{1}$$

where $I$ is the immediate timing (hour-by-hour variation due to changes in prevailing meteorology), $G$ describes the gross seasonal timing (also termed the 'phenology factor') and $S$ provides the spatial source distribution for a given season. The functions $I$, $G$ and $S$ are each dependent on other factors, which may include modelled meteorology, land-use data or satellite data; these details are discussed in subsequent sections.

Table 2 gives the combinations of options for calculating $E(x, y, t)$ that are tested in this study. Each emission methodology

is run for three months between October and December 2017 to cover the period of the pollen measurements. The modelled pollen is also averaged on a 24 hourly basis (to 09:00h each day) to be consistent with the 2017 pollen observations.





### 3.1.1 Immediate timing, $I$

We consider two representations of the immediate timing function ($I$). The first, and simplest assumes that emissions are related to transport and therefore are proportional to the surface wind speeds, used in scenarios E1, E2 and E3. The second method, used in scenarios E4, E5, E6, E7 and E8 accounts for several meteorological factors, treating them as having independent

effects.

$$I(x,y,t) = f_h \cdot f_{\mathrm{RH}} \cdot f_{\mathrm{PR}} \cdot f_{\mathrm{WS}} \cdot f_{\mathrm{TM}} \tag{2}$$

where the terms $f_h$, $f_{\mathrm{RH}}$, $f_{\mathrm{PR}}$, $f_{\mathrm{WS}}$ and $f_{\mathrm{TM}}$ represent, respectively, the response to hour-of-day, RH, precipitation, wind speed and temperature. This approach is similar to Sofiev et al. (2013, eq. 12), representing pollen emissions from birch trees. The assumption is that grass pollen emissions are greatest when conditions are hotter, windier, drier, with less rain, and around

midday. The midday assumption stems from an observational study conducted near Melbourne showing the peak timing of ryegrass pollen release (measured as the number of exposed anthers) occurs during daylight hours and in the early afternoon (Smart and Knox, 1979, fig. 6). This timing is represented as a Gaussian distribution with a mean at the local solar noon (12:00 h) and a standard deviation $\sigma_{\mathrm{h}}$, of either 2 or 4 hours (Smart and Knox, 1979). The larger $\sigma_{\mathrm{h}}$ parameter allows for a wider peak in pollen around noon-time in later scenarios E6, E7 and E8.

For RH we adapt the approach of Sofiev et al. (2013), who used a piece-wise linear relationship scaled from 1 (RH of 50% or less) to 0 (RH of 80% or above). For wind speed, Sofiev et al. (2013) assumed a smaller emission rate ($f_{\mathrm{stagnant}} = 0.33$) in stagnant conditions and scaled smoothly to a saturation value (1.0) for higher wind speeds. We adapt this approach to the case of RH, but use a logistic function ($f_l(y; \alpha, c) = \frac{1}{1+e^{-\alpha(y-c)}}$, for location parameter $c$ and rate parameter $\alpha$), where the rate and location parameters are set to yield $f_l(50; \alpha_{\mathrm{RH}}, c_{\mathrm{RH}}) = 0.95$ and $f_l(80; \alpha_{\mathrm{RH}}, c_{\mathrm{RH}}) = 0.05$, with $\alpha_{\mathrm{RH}}$ being negative, meaning

that the assumed emissions rate decreases with increasing humidity. The final $f_{\mathrm{RH}}$ is then:

$$f_{\mathrm{RH}} = f_{\mathrm{stagnant}} + (1 - f_{\mathrm{stagnant}}) \cdot f_l(\mathrm{RH}; \alpha_{\mathrm{RH}}, c_{\mathrm{RH}}) \tag{3}$$

The equation for the temperature term ($f_{\mathrm{TM}}$) is identical to the RH term (Eq. 3), but taking temperature ($^\circ$C) as the argument and with different rate and location parameters. These are defined such that $f_l(6; \alpha_{\mathrm{TM}}, c_{\mathrm{TM}}) = 0.05$ and $f_l(24; \alpha_{\mathrm{TM}}, c_{\mathrm{TM}}) = 0.95$. The implied rate parameter ($\alpha_{\mathrm{TM}}$) is positive, meaning that grass pollen emissions are assumed to increase with increasing

temperature.

A similar approach is taken for precipitation ($f_{\mathrm{PR}}$), with the logistic rate and location parameters constrained to satisfy $f_l(0; \alpha_{\mathrm{PR}}, c_{\mathrm{PR}}) = 0.95$ and $f_l(0.5; \alpha_{\mathrm{PR}}, c_{\mathrm{PR}}) = 0.05$, where the precipitation is given in units of mm hr$^{-1}$ and $\alpha_{\mathrm{PR}}$ is negative. We cannot impose a constraint of the function being 1.0 for zero precipitation, as the logistic function approaches 1.0 asymptotically. Instead, we scale the result based on the function's value for zero humidity (defined above as 0.95), resulting

in:

$$f_{\mathrm{PR}} = f_{\mathrm{stagnant}} + (1 - f_{\mathrm{stagnant}}) \cdot f_l(\mathrm{PR}; \alpha_{\mathrm{PR}}, c_{\mathrm{PR}}) / f_l(0; \alpha_{\mathrm{PR}}, c_{\mathrm{PR}}) \tag{4}$$





As noted above, the effect from wind speed ($f_{\mathrm{WS}}$) is assumed to scale smoothly from a lower rate $f_{\mathrm{stagnant}}$ in still conditions. We follow the parameterisation of Sofiev et al. (2013, equation 11):

$$f_{\mathrm{WS}} = f_{\mathrm{stagnant}} + (1 - f_{\mathrm{stagnant}}) \cdot (1 - \exp(-\mathrm{WS}/U_{\mathrm{satur}})) \tag{5}$$

where wind speeds ($\mathrm{m\,s^{-1}}$) are scaled by a saturation wind speed ($U_{\mathrm{satur}} = 5\ \mathrm{m\,s^{-1}}$).

### 3.1.2 The gross timing, $G$

We consider two representations of the gross timing, a Gaussian distribution to represent the growth and decline of the spring-time pollen season, and the Enhanced Vegetation Index (EVI). The Gaussian distribution (Eq. 6) is normalised to integrate to the theoretical maximum spatial production of ryegrass pollen over the season, estimated by Smart et al. (1979) as 464 kg ryegrass pollen hectare$^{-1}$ in grasslands to the north of Melbourne.

$$G(x,y,t) = \frac{F}{\sqrt{2\pi\sigma^2}} \exp\left[-\frac{(d-n)^2}{2\sigma^2}\right] \tag{6}$$

where $d$ is the day number (from Oct 1$^{\mathrm{st}}$ to Dec 31$^{\mathrm{st}}$ = 92 days) within the season, $n$ is the mean day number of that season (46.5), $\sigma$ is the standard deviation (26.7), and $F$ is a normalisation factor of $9.53 \times 10^{-8}$, so that seasonal emissions integrate to 464 kg ha$^{-1}$. This Gaussian representation is used in scenarios E1, E5, E6 and E7.

We apply a second Gaussian representation in scenario E8 which uses the shapes of the 2017 observed pollen time-series to 'shift' the distribution by either moving the mean earlier or later in the grass season, and/or adjusting the standard deviation to be tighter or wider. The curves are fitted by optimising the root mean squared error (RMSE) between the pollen counts and the original Gaussian distribution (shown in the supplementary material). The peak of the grass pollen season is earlier in Bendigo and Dookie than day 46.5, thus all grassland north of 37 °S replaces $n$ with 34.7 and $\sigma$ reduces to 15.5 ($F$ remains the same as above). The peaks in observed pollen at Creswick and Churchill are later in the season and count more pollen than other sites, thus at locations south of 37 °S and east of 143.5 °E, $n$ is replaced with 50.5, $\sigma$ is narrowed to 19.3 and $F$ increased to $1.2 \times 10^{-7}$. At sites west of 143.5 °E (i.e. Hamilton), the peak of the pollen observations are greater and distributed more tightly, thus $n$ reverts to 48.1, $\sigma$ is narrowed to 7.7 and $F$ is increased further to $1.56 \times 10^{-7}$.

### 3.1.3 Enhanced Vegetation Index, EVI

Devadas et al. (2018) developed a non-linear statistical model for pollen concentrations using satellite greenness indices across areas surrounding a receptor point. The EVI is a measure of landscape greenness, which is less affected by saturation in higher biomass regions than the widely used Normalised Difference Vegetation Index (Huete et al., 2002). The EVI value typically increases rapidly with time during spring due to foliage growth in deciduous trees or grass growth. In the Victorian temperate climate, fresh grass rapidly dries (or 'cures') in late spring and early summer, causing a fall in EVI. Given the absence of deciduous forests in Australia, most of the temporal variation in EVI is due to grass growth and curing. Here we investigate a relationship between the timing of the pollen season and the gradient in EVI over a region in the South West of Victoria,





spanning 37.3-38.3 °S and 142-143.3 °E (appearing as dashed lines in Figure 3). This region is upwind of Melbourne, in terms of the prevailing climatological wind, and has high agricultural activity.

Using the Moderate Resolution Imaging Spectroradiometer (MODIS) MOD13C1 data (from the Terra satellite and at 0.05° resolution), Figure 2a shows the gradient in averaged EVI drops off rapidly, around the same time as the pollen season peaks.

Figure 2b shows the first derivative of EVI is anti-correlated with the grass pollen time-series at UoM. If we examine interannual variation, assessing the day-of-year when the EVI falls most rapidly (represented as the middle of the 16-day EVI compositing window) and the day-of-year when the grass pollen peaks (having first applied a smoothing spline to the pollen time-series), a relationship between these two quantities is observed: the Pearson correlation is 0.4, the slope of the linear regression is 1.006 and the means of the two Julian dates differs by only 2.7 days (Figure 2c). This agreement is especially notable given the

uncertainty induced by the wide EVI compositing window.

Taking this one step further, we apply a similar analysis to each individual $0.05 \times 0.05°$ MODIS pixel (Figure 3). Given the high deposition velocity of grass pollen grains (4.6 cm s$^{-1}$, as discussed above), the contribution of pollen emitted from the productive grassland areas in western Victoria to observations recorded in Melbourne is likely to be minimal. However, this analysis may help inform our understanding about the relationship between the remotely-sensed vegetation index and

broad-scale features of the pollen season. The timing of the fall in EVI in South West Victoria not only correlates well with the timing of the grass pollen season experienced in Melbourne (Figure 3a), but also the differences in timing are relatively small (Figure 3b). The north-west of the state is generally much drier than in the south-east (Figure 1c), and the north-west area dries out earlier in the year (Figure 3c). Areas identified as crops or pasture (Figure 1d) demonstrate a more rapid fall in EVI (Figure 3d).

This exploratory analysis suggests that in this bioclimate the broad parameters of the pollen season can be diagnosed from the EVI fields. On a broad temporal scale, a fall in EVI over pollen source regions is associated with increasing pollen emissions. In light of this, we consider an EVI-based representation of the gross timing ($G$).

$$G(x,y,t) = \max\left(0, -\frac{\partial \mathrm{EVI}(x,y,t)}{\partial t}\right) \tag{7}$$

The $\max(\cdot)$ function ensures that the emissions are strictly positive. We note that Eq. 7 incorporates both temporal and spatial

information, and can thus be used to represent the spatial distribution ($S$), in which case we can set $S(x,y) = 1.0$ for all grid-points $(x,y)$ (scenario E3). Alternatively, we can use the same spatial forcing (based on an assumed land-use classification) to provide an extra spatial constraint. $\partial \mathrm{EVI}$ is used in scenarios E2 and E4.

### 3.1.4   The spatial function, $S$

Mapped grass and pasture for Victoria were extracted from the Australian Land Use and Management (ALUM) Classification

(ABARES, 2017) and were re-gridded from 50 m resolution to the 3 km grid used by the C-CTM. ALUM includes 193 categories of which only three are assumed to overlap with grazing pastures ('Grazing modified pastures', 'Native/exotic pasture mosaic' and 'Grazing irrigated modified pastures'); the fractional coverage of these three classes together is shown in Figure 1d. While many cultivated cropping cereals grown in the region are also grasses (e.g. wheat, barley), they are mostly





self-pollinating and thus produce very little pollen compared to wind-pollinated grass species such as ryegrass. The area to the east of Melbourne is mountainous and therefore not arable (Figure 1e), whilst the region to the northwest is arid (Figure 1c). The most productive areas of pasture grass in Victoria are found in the west of the region near Hamilton and south west of Churchill. The ALUM grass map is used in scenarios E1, E2, E4, E5, E6, E7 and E8.

### 3.1.5 Pollen production and loss model

In reality, there is a finite amount of grass pollen available for release at a given time, and once exhausted (by dry or wet deposition or advection) pollen is only replenished at a finite rate. We develop a 'production-loss' model for scenario E7.

$$E(x,y,t) = A(x,y,t) \cdot I(x,y,t) \tag{8}$$

$$A(x,y,t) = A(x,y,t-\delta t) + P(x,y,t-\delta t) - L(x,y,t-\delta t) \tag{9}$$

$$P(x,y,t) = S(x,y,t) \cdot G(x,y,t) \cdot \left(\frac{\delta t}{T}\right) \tag{10}$$

$$L(x,y,t) = A(x,y,t) \cdot \exp(-\lambda \cdot \delta t) \tag{11}$$

where $A$ is the amount of pollen available for release on the grass at grid-point $(x,y)$ at time $t$, $\delta t$ is the model time-step, $P$ is the amount produced on the plant between $t$ and $t+\delta t$, $L$ is the amount lost between $t$ and $t+\delta t$, $T$ is the total length of the grass pollen season and $\lambda$ is the loss rate (due to direct dry or wet deposition). We can interpret the above as follows; emissions are set to be the product of the available pollen load and the instantaneous emission factor (Eq. 8). The available pollen load is the sum of available pollen at the last time-step and pollen produced since then, less any loss since the last time-step (Eq. 9). The pollen produced is given by the product of the spatial and gross-timing terms, proportional to the fraction of the grass pollen season covered during the time-step (Eq. 10). The pollen lost is based on an exponential decay (Eq. 11), assumed to incorporate direct wet and dry deposition before the pollen leaves the grid-cell. The loss decay parameter ($\lambda$), is defined as a piece-wise polynomial function based on the rain rate, such that pollen has a half-life on the plant of two days in dry conditions and 12 hours in wet conditions, with the latter corresponding to a rain rate of $2\,\text{mm}\,\text{h}^{-1}$.

### 3.1.6 Statistical Models

In parallel to the emission-dispersion modelling presented here, statistical forecasting methods have been trialled for use in Victoria. These models are non-linear regression equations that use weather model data, derived parameters from the MODIS EVI, and land-use maps as predictors. These data can be decomposed into a slow-moving seasonal component (similar to gross-timing term described above) and a second component that accounts for day-to-day variation. The models were trained on daily pollen count data, and thus cannot resolve higher-resolution temporal variation. The two statistical models are described in detail in Silver et al. (2019), and summarised here. 'V1' used data from Melbourne spanning 2000-2016 (scenario E9) while 'V2' also used the 2017 data from the eight Victorian sites (scenario E10). The V1 model was developed ahead of the 2017 pollen season before counts were available at the new pollen sites, as the BoM required input for their pilot thunderstorm asthma service. The seasonal component was represented as a Cauchy distribution (which decays more slowly than a Gaussian





distribution), with a fixed scale parameter ($k = 19$ days). The magnitude of the pollen season (corresponding to the maximum of the seasonal term) was estimated by univariate linear regression on the winter-time maximum EVI. The timing of the seasonal maximum was estimated by the day-of-year when the EVI falls to 0.05 below its winter-time maximum. The magnitude and timing were smoothed spatially using an inverse cubed distance weighting.

Both V1 and V2 were constructed as Generalised Additive Models (Wood, 2006), a form of multivariate regression that allows for a non-linear influence of the predictor variable on the response variable. The response variable used was the $\log(x+1)$-transformed pollen count. The log of the Cauchy term and a number of derived weather parameters were considered for inclusion in the model. Each model was built up via forward step-wise variable-selection: starting with a "null model" (predicting nothing but the mean), terms were considered for inclusion. Each predictor was trialled as having a linear

or alternatively non-linear effect on the response variable, and the out-of-sample prediction skill was tested. The combination of predictor and form (i.e., linear or non-linear) that yielded the biggest gain in predictive skill was retained. This procedure was repeated until the incremental impact on predictive skill of additional terms was negligible. The model skill was tested by leaving out entire pollen seasons, fitting the model without these data, then assessing on the out-of-sample subset. Model skill was quantified using the Pearson correlation between predicted and observed pollen.

The statistical models were adapted for 3D dispersion modelling to use hourly meteorological inputs (or daily, in the case of precipitation). The adapted forms of the two models are:

$$
\begin{aligned}
\log(1 + \mathrm{P}_1(x,y,t)) \;=\; & -0.290 + 0.970 \cdot \mathrm{R}_1(x,y,d) - 0.183 \cdot \log(\mathrm{PR}(x,y,d)+1) - 0.117 \cdot \log(\mathrm{PR}(x,y,d)) \\
& + \; f_{\mathrm{TM1}}(x,y,t) + f_{\mathrm{RH1}}(x,y,t) \qquad\qquad\qquad\qquad\qquad\qquad\qquad\qquad (12)
\end{aligned}
$$

$$
\begin{aligned}
\log(1 + \mathrm{P}_2(x,y,t)) \;=\; & 1.225 + 0.770 \cdot \mathrm{R}_2(x,y,d) - 0.033 \cdot \mathrm{WS}(x,y,t) + f_{\mathrm{RH2}}(x,y,h) \\
& + \; f_{\mathrm{TM2}}(x,y,t) + f_{\mathrm{PR}}(x,y,d) \qquad\qquad\qquad\qquad\qquad\qquad\qquad\qquad\;\; (13)
\end{aligned}
$$

where $P$ is the predicted pollen emission for version $i$ on at grid-point $(x,y)$ and time $t$, $\mathrm{R}_i$ is the seasonal term based on the EVI parameters at grid-point $(x,y)$ and for day $d$ (outlined below) and *WS* is the wind-speed (m s$^{-1}$). In both versions of the statistical model, the variable-selection process assigned a non-linear response to the temperature, $f_{\mathrm{TMi}}$ ($^\circ$C), and RH $f_{\mathrm{RHi}}$ (%). Only V2 uses a non linear term for daily precipitation, $f_{\mathrm{PR}}$ (mm). The non-linear relationships between pollen emission

and increasing temperature, RH and precipitation are shown in Figure 4.

For both V1 and V2, the seasonal term contributes the most variance to the modelled emissions followed by the temperature term. The temperature term in both models was associated with a strong (near-linear increase) until a value of 25-30 $^\circ$C, after which there is a rapid decline (Figure 4a,c). This decline in concentrations may be due to increased boundary layer heights (and thus greater effective dilution) rather than a decrease in emissions.





The seasonal term based on the EVI parameters is given as:

$$R_i(x,y,d) = \log(SF_i(x,y) \cdot f_C(d, \mu_i(x,y), k)), \text{ where} \tag{14}$$

$$f_C(d, \mu, k) = \left( \pi \cdot k \cdot \left[ 1 + \left( \frac{d - \mu}{k} \right)^2 \right] \right)^{-1} \tag{15}$$

$$SF_1(x,y) = \max(-4355.913 + 21490.343 \cdot E_{\text{max,smoothed}}(x,y), 10^{-10}) \tag{16}$$

$$\mu_1(x,y) = E_{\text{drop,smoothed}}(x,y) \tag{17}$$

$$SF_2(x,y) = 267.627 + 8853.990 \cdot E_{\text{max,smoothed}}(x,y) \tag{18}$$

$$\mu_2(x,y) = 202.478 + 0.385 \cdot E_{\text{drop,smoothed}}(x,y) \tag{19}$$

where the scale factor parameter $SF_i(x,y)$ is based on the smoothed value of the winter-time maximum EVI ($E_{\text{max,smoothed}}(x,y)$), while the timing of the peak of the pollen season ($\mu_i(x,y)$) was assumed to scale linearly with smoothed field of the day-of-year when the EVI drops 0.05 below its winter-time maximum (see Silver et al., 2019, for further details).

## 3.2 Statistical evaluation

The skill of the pollen forecasts depends in part on how well the meteorology is predicted. The Pearson correlation indicates the strength of the correspondence without consideration of differences in magnitude, whist the index of agreement (IOA, described in the supplementary material) is a good indicator of model performance. The normalised mean bias (NMB) gives the relative difference between the model and observations.

To determine the best pollen emission methodology, we look for skill in the ability of VGPEM to forecast the possibility of the pollen being classed as high or extreme (> 50 grains m$^{-3}$), which is a level that health impacts may be felt more strongly. The number and timing of predicted high pollen days is evaluated quantitatively for consistency and accuracy, by calculating the probability of detection (POD), the false alarm ratio (FAR), and the equitable threat score ( ETS) from a simple table of model outcomes (Table 3). The POD is the fraction of correctly identified high model forecasts compared to the observations, between 0 and 1.

$$POD = \frac{a}{a + c} \tag{20}$$

The FAR puts a value between 0 and 1 on how many of the predicted high pollen days did not correspond with an observed high pollen day.

$$FAR = \frac{b}{a + b} \tag{21}$$

The ETS is the fraction of modelled high pollen days that were correctly predicted, and adjusted for correctly modelled days occurring with random chance. The ETS value is between -1/3 and 1, with a score of 0 indicating no skill; this is defined as

$$ETS = \frac{a - a_{\text{random}}}{a + b + c - a_{\text{random}}}, \tag{22}$$

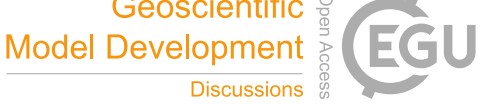



where

$$a_{\mathrm{random}} = \frac{(a+c) \times (a+b)}{a+b} \tag{23}$$

As Zink et al. (2013) point out, low skill scores are given to models whose pollen concentrations are close to observed concentrations yet fall into separate 'risk' categories. For example the model predicts 48 grains m$^{-3}$ and classed as 'moderate' whilst the observations are 52 grains m$^{-3}$ and are 'high'. Therefore we also evaluate the modelled pollen against the observations in terms of their Pearson correlation, RMSE and Gerrity score. The Gerrity score puts a value on the accuracy of VGPEM in predicting all the observed pollen categories, relative to that of random chance (Gerrity, 1992). Gerrity scores range between -1 and 1, with 0 indicating no skill and 1 being a perfect model. Calculation of Gerrity scores is complex and is described fully in the supplementary material.

The best forecasting methodology will have a high Pearson correlation, Gerrity score, POD and ETS and a low FAR and small RMSE.

## 4 Results and discussion

### 4.1 Verification of meteorology

Meteorological variables are extracted from the ACCESS runs at the locations of the AWS closest to the pollen observation sites (Table 1). At some pollen observation sites the AWS are located more than 10 km away, and nearly 30 km away in the case of Dookie. A direct comparison is made of hourly temperature, wind speed, wind vectors, precipitation and RH between ACCESS and the AWS observations, using the Pearson correlation, IOA and NMB. (Figure 5). Here the NMB is normalised by the mean of the absolute value of the observations (as opposed to the mean of the observations) because wind vectors contain negative values. Temperature and RH are both modelled with a high degree of accuracy at all sites, demonstrating a high Pearson correlation (average r=0.9), almost no bias, and high IOA (average IOA=0.8). Predicted wind speeds are biased slightly low (average NMB=-0.2). The $V$ (north–south) component is similarly well modelled as the $U$ (east-west) component (average $V$ r=0.80 compared to average $U$ r=0.77). Precipitation has a low degree of bias (average NMB=0.17) but not particularly well correlated with observations (average r=0.21), and has a lower overall IOA (average IOA=0.55).

### 4.2 Observed and modelled pollen correlations with meteorology

We assess which measured AWS meteorological variables are most strongly related to the observed pollen. Figure 6a shows observed grass pollen is most strongly correlated with temperature at the majority of sites (average r=0.44), and most negatively correlated with RH (average r=-0.34).

Observed wind speed is not strongly related to observed grass pollen, except when combined with direction, specifically the $U$ wind vector is generally a stronger predictor of pollen (average r=0.32) than the $V$ wind vector (average r=0.22). Precipitation washes pollen from the air, but shows no correlation here as rain during the 2017 season was infrequent (average r=0). Pollen



observations at Dookie are the least correlated with any of the meteorological variables, perhaps because the closest AWS is 29 km away.

Figure 6b shows Pearson correlations for the modelled pollen against ACCESS meteorology, using scenario E8 as an example that uses the meteorological timing function. The strengths of the modelled correlations are broadly similar to those observed in Figure 6a, but the model is more strongly coupled to wind speed (average r=0.25) and less correlated to the $U$ wind vector than is observed (average r=-0.07). Transport of pollen from the productive grasslands in the west of Victoria to Melbourne would rely on the $U$ wind vector being modelled accurately, however the model lifetime of these pollen grains is 6 hours over a height of 1 km; too short for pollen emitted near Hamilton to reach Melbourne. We extracted the boundary layer height from the model (unavailable in the observations), which showed that the modelled grass pollen is more strongly correlated to atmospheric dilution (average r=0.61) than it is to temperature (average r=0.44). The model RH is more negatively correlated with grass pollen levels (average r=-0.52) than is observed.

### 4.3 Verification of pollen source methodologies

The modelled pollen concentrations are first normalised by the observed seasonal mean across all observation sites, equal to 47 grains m$^{-3}$. This normalisation allows the evaluation of trends in the daily grass pollen concentrations without considering their magnitude, as this can be corrected later. For 2017, observed individual site means range from 31 grains m$^{-3}$ at Melbourne to 60 grains m$^{-3}$ at Creswick. The lowest means are found in the densely populated regions of Geelong, Melbourne and Burwood (see figure 1f). Figure 7 shows correlations and statistical results for each pollen observation site. Numbers of observed days in the lumped high and extreme category (> 50 grains m$^{-3}$) are above 20 days for all sites. The points are coloured red for Gaussian emissions methodologies, yellow for $\partial$EVI, green for the production and loss model and blue for statistical methodologies describing the pollen season.

E1, E2 and E3 used wind speed as the immediate timing function, which did not provide good prediction skill scores (average r=0.25, 0.18 and 0.17 respectively). Observed wind speed was not strongly correlated to observed pollen, but it was useful to test this parameter in Victoria. Subsequent method E5 used the meteorological timing function that included temperature and RH and performed better (average r=0.43). Widening the timing of the peak pollen emission from 2 to 4 hours, as included in E6, improved results further over E5 (average r=0.44).

At most sites the Gaussian description of the season performed better than the $\partial$EVI, shown by improvements in FAR of E1 over E2 both of which used wind speed as the immediate timing descriptor (average FAR=0.57 and 0.61 respectively), and E5 over E4 both of which used the meteorological timing function (average FAR=0.48 and 0.52 respectively). These results point to using $\partial$EVI data as a descriptor of the pollen season as providing poor skill at most of the sites. The $\partial$EVI data is very noisy. However E4 using the $\partial$EVI data and a meteorological timing function gives a good Gerrity score (0.54) and high POD (0.92) and ETS (0.40) at Dookie. When other elements of the EVI data are used in the statistical models (E9 and E10), such as the winter maximum and the day on which the EVI falls below 0.05 of the winter maximum, the pollen prediction is much improved at most sites (average POD E9=0.67 and E10 0.69).

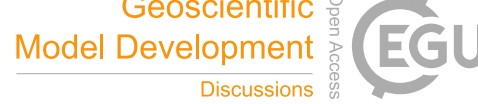



The pollen production and loss model E7 had a very high POD (0.96) at Hamilton, but the method was less effective elsewhere with high FAR and RMSE scores at the other observation sites (average FAR=0.55 and RMSE=63). E7 used the Gaussian distribution for the seasonal term which could be improved upon, but the method was superseded by the good performance of the statistical models (with the exception of E9 in Hamilton and Geelong). The performance of E10 indicated

improvements in forecasting skill at all sites with the exceptions of Dookie and perhaps Bendigo. E10 predicted the lowest FAR of high pollen predictions at five of the eight observation sites (average FAR=0.37). The ETS adjusts the model score for achieving high pollen predictions at random. E10 achieves higher ETS scores at four of the eight sites (average ETS=0.35).

The Geelong pollen observations are not well modelled by most of the emission methodologies with Gerrity scores mainly in the negative region and high FAR greater than 0.8. E10 provides the best scores by far at Geelong with a 0.62 Pearson

correlation and 0.3 Gerrity score, though overall, results are poorer for Geelong than any other site.

If the best performing scenario for each observation site and under all scoring methods is counted from Figure 7, shifted Gaussian methodology E8 is best 12 times, statistical representation E9 is best 4 times and E10 25 times. E9, built using data prior to the 2017 season, has a stronger dependence on precipitation than E10, which is not supported by the correlation of 2017 pollen with meteorology. This suggests that the V2 statistical approach to the immediate timing combined with an EVI-based

approach to the gross timing and a spatial source, is likely to produce the most accurate pollen forecasts.

It is useful to plot the observed and modelled pollen as a cumulative time series, as this indicates the timing of increased and decreased pollen counts (Figure 8). Here we focus on the best performing scenarios from each of the seasonal emission methods, capturing the range in descriptions of the pollen season. The observations show an "S" shaped profile, with increased pollen gradients in November. By the end of the 2017 season, all modelled profiles reach a cumulative total of 4200 grains m$^{-3}$

due to normalisation to the same observed mean value.

The $\partial$EVI method E4 tends to emit grass pollen too early in the season compared with observations at most sites. However at Dookie the shape of the season in E4 and E8 compares better to the observations and E4 captures the mid November change when pollen counts decrease, better than E10, The observed pollen at Dookie experienced a much larger grass pollen input from the middle of October to early November than E10 (but is represented well by E8). There is little additional observed

pollen at Dookie after early November, which is at least 20 days earlier than at the other sites. At Melbourne, Bendigo and Burwood, both E8 and E10 predict the early part of the grass pollen season very well, but emitted too much modelled pollen towards the end of the season. The steeper gradient in E8 and E10 at Melbourne between the middle to the end of November shows that too much pollen was being emitted then. In contrast, observed pollen at Churchill, Creswick, and Hamilton show a rapid increase in emissions at the end of November which is not matched by either E8 or E10. However the observations

suggest that additional pollen continues to be emitted towards the end of December at Creswick and Churchill, prolonging the season. Scenario E8 captures the steep November gradient at Hamilton very well.

The observed and modelled pollen cumulative profiles in Melbourne, Geelong and Burwood are less smooth than the other regional sites, perhaps indicating more atmospheric variability near the coasts. It might also indicate the larger distances between the urban sites and the grass pollen production regions (more transport, less local production), as compared to the





monitoring sites within grass pollen production areas (less transport, more local production). Here we also note the apparent lack of an 'S' shape in the modelled profiles at Geelong, which may account for the poor model performance at this site.

Table 4 splits the best model predictions from E8 and E10 into low, moderate, high and extreme categories to directly compare with observed categories. Here data from all counting sites are combined to ensure a large sample size. The diagonal

in each table highlights the number of days the model has correctly predicted the observed category. Values far from the diagonal indicate the model has under or over-predicted the observed pollen. We want to avoid occasions where the observed pollen is extreme, but the model predicts low pollen. Table 4 shows that both E8 and E10 have good skill in predicting low observed pollen days. Both models also show high occurrences of predicting moderate pollen when the observed category is low. Siljamo et al. (2013) found difficulties in modelling moderate category days, which is not the case here. Both models are

equally good at predicting the high and extreme observed categories. E10 has less occurrences than E8 of predicting extreme pollen on days when observations were low. However there are six cases in E8 and one in E10 that predict low pollen when the observations were extreme. These cases occur around the 10-11$^{\text{th}}$ November within the city at Geelong, Melbourne and Burwood. Examining the meteorology from this period (pollen counts are date-stamped at 9am, but represent the preceding 24 hours), shows the model has captured the observed temperature, wind speed, direction and zero rainfall. The observed wind

direction is from the south and south east, bringing mainly clean, marine air. However observed pollen is extreme on these days, suggesting a highly localised source. One explanation is that only pasture grass is considered in the model, whereas grass is usually present in most other land-use categories. There is green space within most cities on both public and private land, and grass plants are efficient at colonising disturbed areas such as road verges. Future development of VGPEM could consider the sub-gridscale grass fraction using high-resolution satellite data sources.

## 20  5  Conclusions

The aim of this work was to develop and assess the utility of a grass pollen emission methodology for use in a pollen forecasting tool for Victoria in Australia. Our work is the first of its kind for Australia, and whilst initially based in the State of Victoria, future work will see the methodology applied nationally.

Grass pollen was observed during 2017 at eight sites in Victoria, showing strongest correlations with temperature (positive)

and RH (negative). Correlations of grass pollen with wind speed and precipitation were not strong.

Ten grass pollen emission source methodologies were presented in this work. Most used the locations of pasture grass in Victoria in combination with meteorological parameters and a seasonal pollen emission parameter. The seasonal parameter was either based on a simple Gaussian representation of time variation, or on the Enhanced Vegetation Index which measures greening from satellite. Each source methodology was run using a host transport model driven by ACCESS numerical weather

predictions, at a spatial resolution of 3 km. The pollen was treated as an inert particle of diameter 35 $\mu$m and 1000 kg m$^{-3}$ density, however these parameters are uncertain and impact the aerodynamic properties of the pollen.

Comparison of predicted meteorology with observations showed that ACCESS is very good at predicting temperature but less so for precipitation, compared to other meteorological parameters. Wind speeds are biased a little low, but are not the





strongest correlating meteorological parameter for observed pollen. Use of wind speed as the immediate timing function in the pollen emissions framework also performed poorly. The key to predictive skill in immediate timing was to use a meteorological timing function that incorporates the parameters most correlated with observed pollen, namely temperature and RH.

Grass pollen source terms using the $\partial$EVI data did not perform particularly well, the Victorian grass pollen season perhaps better described by a simple Gaussian variation. Whilst emission method E8 worked well, the method is limited by fixed timings for the start and end of the pollen season, and the distributions were only trained on the 2017 Victorian observation data. These data may not account for regional variation nor inter-annual variability.

Implementing the maximum EVI value together with the date on which the EVI falls to 0.05 below this maximum, within a statistical methodology predicted pollen concentrations with much better skill. The statistical model E10 uses these EVI data together with 25 years of observational data from the UoM, and one year from the seven other Victorian sites. As the EVI comes from satellite data and varies spatially and temporally, these methods are more suitable for future years, and have potential to work well outside of Victoria. The E10 methodology will be implemented in VGPEM1.0.

Long term observations are vital to record the grass pollen emission strength across Victoria in future years, particularly tracking changes brought about by climate change, changes to agricultural practices and the growth of cities into rural areas. The new Victorian pollen observation stations established after the thunderstorm asthma event in November 2016 should be maintained to aid forecasting of potential threats in future. Advances in technology may provide automated pollen counting in future which would improve the temporal resolution, and the possibility of recognising ruptured pollen grains. These technologies are required to support pollen forecasting, and to constrain future modelling of the pollen rupturing process.

*Code availability.* The pollen emissions code is available as a text file in the supplementary section.

*Data availability.* The Victorian pollen counts and forecasts from all eight sites are disseminated to the public via the web, a smartphone app (named "Melbourne Pollen Count" for iOS and Android) and an automated Facebook and Twitter account (@MelbournePollen).

*Author contributions.* KME and JDS devised the experiments, wrote the VGPEM code and most of the manuscript. KME ran the C-CTM. EN and ERL oversee six of the Victorian pollen count sites. CS oversees the two Deakin pollen count sites. AW provided the ACCESS meteorology. EE is the project manager. All authors edited the manuscript.

*Competing interests.* The authors declare that they have no conflicts of interest.





*Acknowledgements.* This work was funded by the Victorian Department of Health and Human Services. We are grateful to the pollen counters at each of the sites. Dr Penelope Jones at the University of Tasmania performed an external audit of the count data examined here. We acknowledge the valuable and ongoing technical support from the University of Melbourne Science IT Team, especially Ms Usha Nattala and Dr Uli Felzmann as well as the team at Infrastructure Services. J. Silver's work on the initial development of the emission module code
5 was funded by a McKenzie Fellowship from the University of Melbourne.



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



**Table 1.** Locations of Burkard pollen sampling network in Victoria arranged west to east, and their nearest Automatic Weather Station (AWS). Code refers to site names within figures of this paper. UoM = University of Melbourne.

| Site | Code | Lon °E | Lat °N | Location | Closest AWS (distance, km) |
|---|---|---|---|---|---|
| Hamilton | H | 142.03 | -37.74 | Hamilton hospital grounds | Hamilton airport (10.4 km) |
| Creswick | Cw | 143.90 | -37.42 | UoM satellite campus | Ballarat aerodrome (15.3 km) |
| Bendigo | Bg | 144.30 | -36.78 | Latrobe University satellite campus | Bendigo airport (5.5 km) |
| Geelong | G | 144.36 | -38.14 | Deakin University Waurn Ponds campus | Geelong racecourse (7.0 km) |
| Melbourne | M | 144.96 | -37.80 | UoM city campus | Melbourne Olympic park (3.3 km) |
| Burwood | Bu | 145.12 | -37.85 | Deakin University Burwood campus | Scoresby (11.4 km) |
| Dookie | D | 145.71 | -36.38 | UoM satellite campus | Shepparton airport (29.3 km) |
| Churchill | Ch | 146.43 | -38.31 | Federation University campus | Latrobe Valley airport (11.6 km) |



**Table 2.** Options tested for pollen emission in this study. EVI = Enhanced Vegetation Index.

| Scenario | Immediate timing, I | Gross timing, G | Spatial function, S |
|----------|---------------------|-----------------|---------------------|
| E1 | Wind speed | Gaussian | Grass map |
| E2 | Wind speed | $\partial$EVI | Grass map |
| E3 | Wind speed | $\partial$EVI | 1.0 (embodied in $\partial$EVI value) |
| E4 | Meteorological function, $\sigma_h = 2$ | $\partial$EVI | Grass map |
| E5 | Meteorological function, $\sigma_h = 2$ | Gaussian | Grass map |
| E6 | Meteorological function, $\sigma_h = 4$ | Gaussian | Grass map |
| E7 | Meteorological function, $\sigma_h = 4$ | Gaussian | Production/Loss model |
| E8 | Meteorological function, $\sigma_h = 4$ | Shifted Gaussian | Grass map |
| E9 | Statistical model V1 | EVI based | EVI based |
| E10 | Statistical model V2 | EVI based | EVI based |



**Table 3.** The $2 \times 2$ contingency table describing each model outcome.

| Model | Observation | |
|---|---|---|
| | Yes | No |
| Yes | $(a)$ Hit | $(b)$ False alarm |
| No | $(c)$ Miss | $(d)$ Correct negative |



**Table 4.** The number of days the model predicts a particular observed pollen category for E8 (left) and E10 (right). Data from all Victorian sites are combined. Coloured text highlights where the model captures the correct observed category

| E8 | Observation | | | | E10 | Observation | | | |
|---|---|---|---|---|---|---|---|---|---|
| Model | Low | Moderate | High | Extreme | Model | Low | Moderate | High | Extreme |
| Low | **229** | 49 | 18 | 6 | Low | **116** | 15 | 4 | 1 |
| Moderate | 69 | **33** | 29 | 13 | Moderate | 189 | **72** | 40 | 17 |
| High | 34 | 32 | **35** | 35 | High | 33 | 30 | **46** | 39 |
| Extreme | 14 | 15 | 21 | **55** | Extreme | 1 | 11 | 10 | **50** |





**Figure 1.** Maps of (a) Victoria within Australia, (b) pollen observing sites in the domain, (c) mean annual rainfall, (d) pasture grass coverage, (e) terrain, (f) population density. Data sources: (c) BoM, (d) ABARES, (e) Geoscience Australia, (f) Bureau of Statistics. The pollen counting site locations are shown.





**Figure 2.** (a) Time series of EVI from South West Victoria (averaged over the region $37.3 - 38.3°$S and $142 - 143.3°$E, shown in each panel of Figure 3) and the grass pollen record in Melbourne. The EVI data are 16-day composites. (b) As panel (a) except presenting the derivative of the 16-day EVI with respect to time. (c) the day-of-year of the minimum of the $\frac{\partial \text{EVI}}{\partial t}$ for each year plotted against the day-of-year of the maximum pollen; when assessing the timing of the pollen grass pollen peak, the grass pollen time-series was smoothed using a cubic smoothing spline. The dashed lines in (c) represent 16 days either side of a given day, which is the width of the MODIS EVI compositing window.



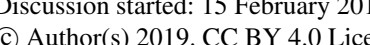


**Figure 3.** Relationship between the timing of the peak in grass pollen in Melbourne with the timing of the sharpest drop in EVI at each MODIS pixel: correlation (a) and the root mean-squared error in the timing (b). Also shown are the average timing (c) and rate (d) of the fastest fall in EVI at each point in the domain. The dashed rectangle in South West Victoria (spanning 37.3-38.3 °S and 142-143.3 °E) displays the region over which the EVI time-series were averaged for Figure 2.





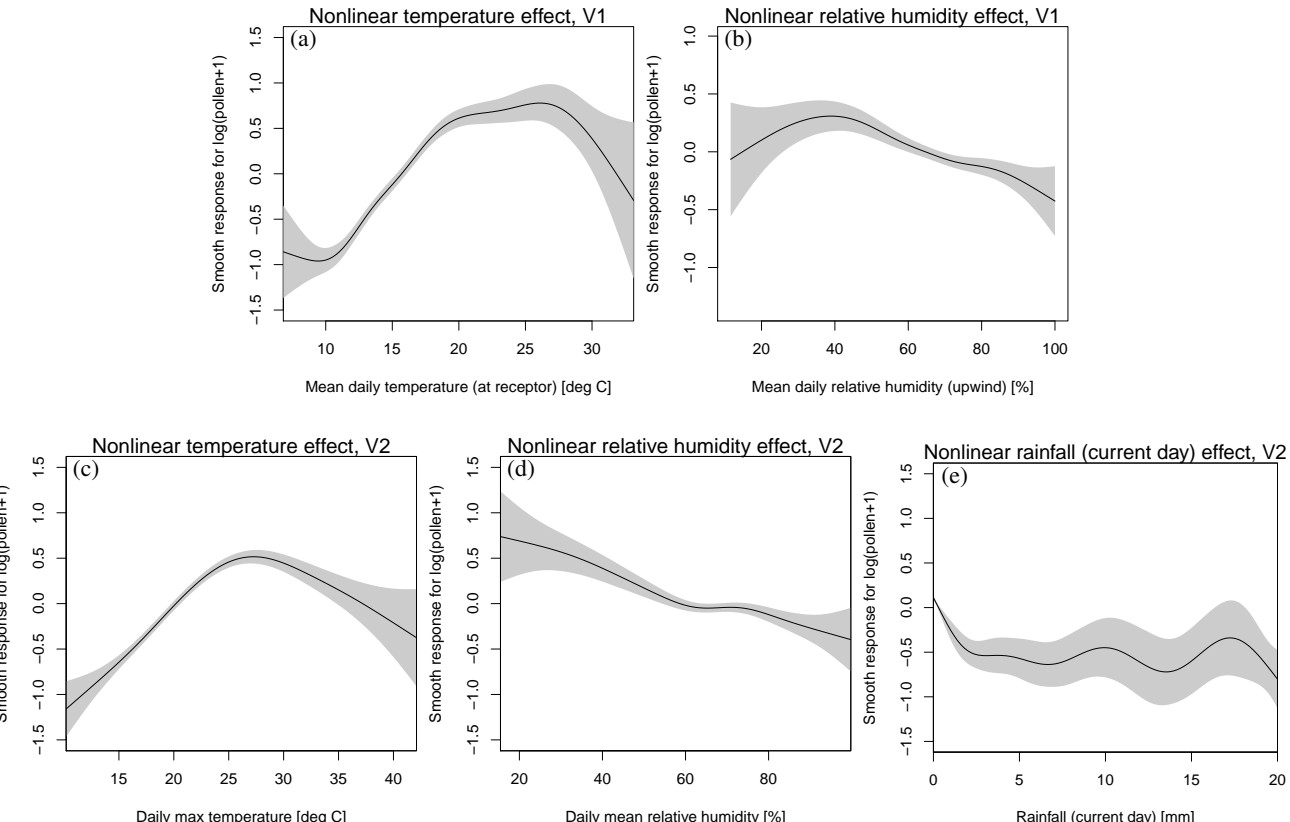

**Figure 4.** The shape of the non-linear terms in the statistical models related to temperature (a and c), relative humidity (b and d) and rainfall (e) for V1 (top row) and V2 (bottom row).



**Figure 5.** Comparison of observed and modelled meteorological variables at Automatic Weather Stations sites nearest to the pollen observation sites. (a) Pearson correlation. (b) index of agreement. (c) normalised mean bias.



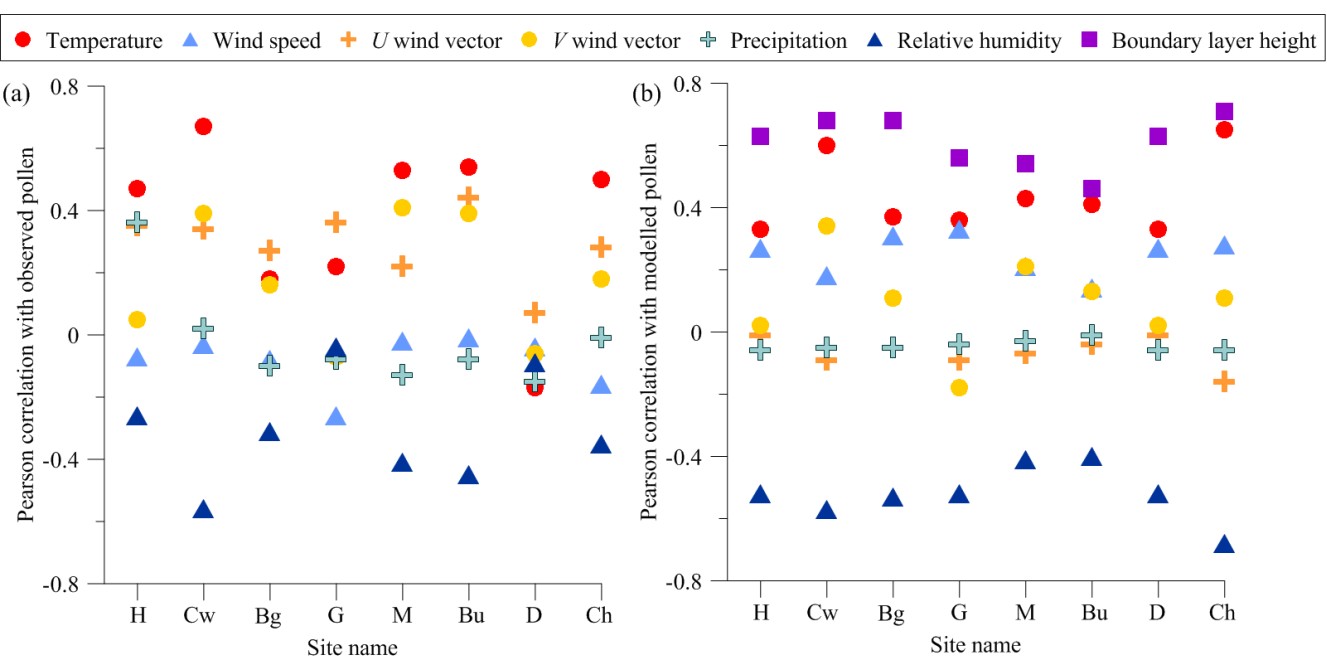

**Figure 6.** Pearson correlations of (a) observed pollen with observed meteorological variables from nearest Automatic Weather Station and (b) modelled pollen with modelled meteorological variables.



**Figure 7.** Results from the pollen emission methodology scenarios (a) Pearson correlation (b) Gerrity score (c) POD (d) ETS (e) FAR (f) RMSE. The sites are presented from west to east, and coloured red relating to Gaussian methodologies, yellow for ∂EVI methodologies, green for the production and loss model and blue for statistical methodologies. A higher score is better for the Pearson correlation, Gerrity score, POD and ETS. A lower score is better for the FAR and RMSE.



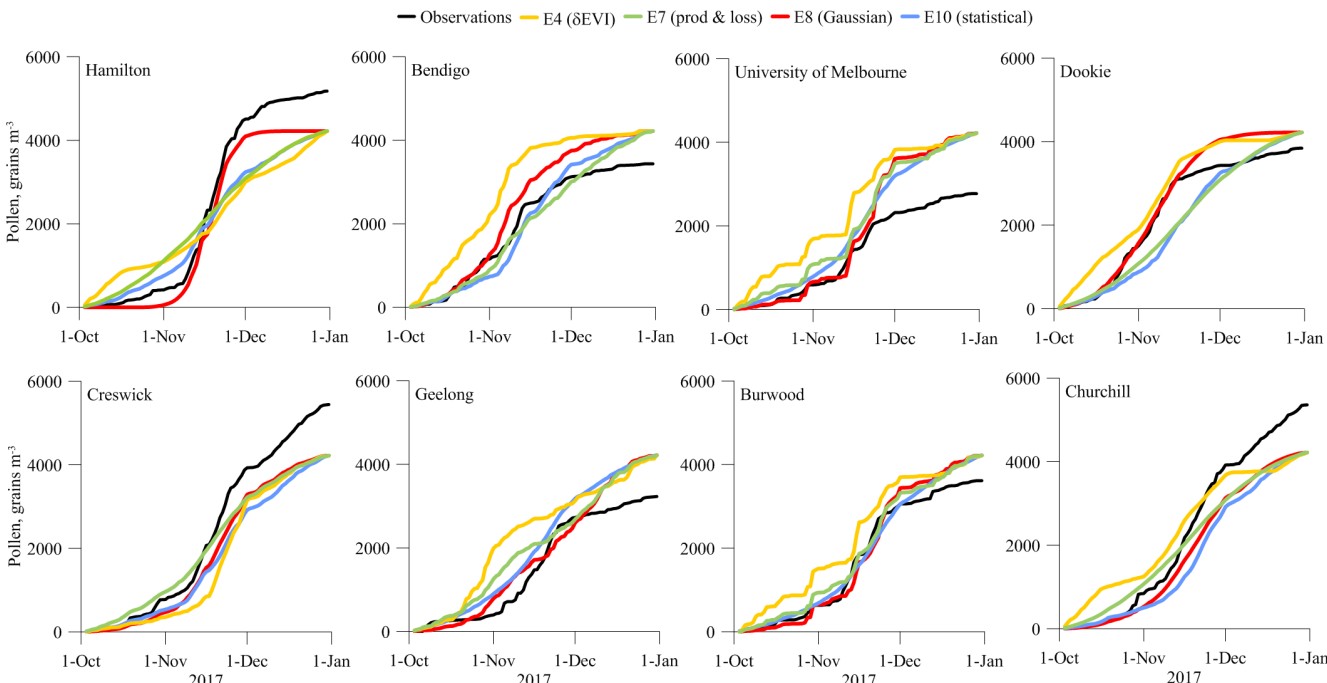

**Figure 8.** Cumulative time series in pollen across the 2017 season. Sites arranged from west to east.