# Peer review of "Development and evaluation of pollen source methodologies for the Victorian Grass Pollen Emissions Module VGPEM1.0"

_Geoscientific Model Development, 2019_

## Referee Comment (RC1) · Anonymous Referee #1 · 28 Mar 2019

General comments: This manuscript describes the implementation of several different emissions parameterizations to prognostically simulate grass pollen in one region of Australia. The paper is relatively narrow in scope in that it is focused on one type of pollen (grass) in one small region of the globe. That said, it will be useful model for the region and has the potential to be applied to other regions for grass pollen. My comments are mostly minor and regarding changes for clarity and additional supporting information. I recommend the article for publication following these revisions.

Comments on the analysis

1. Regarding the statistical model functions shown in Figure 4: It is helpful to see what

the dependencies look like, but it is unclear how these meteorological factors physically relate to what we know about pollen emissions. Can these observed dependencies be related to any physical processes?

2. Risk category definitions: Page 4 – line 14 – Can you place these count categories (low, moderate, high, extreme) in context as compared to other counts in other regions (e.g., Europe, US)? This would give readers an idea of how high/low Australian counts are relative to other locations in the world. Additionally, for the evaluation based on risk categories on page 12, how do uncertainties in these thresholds influence the analysis?

3. Page 6 line 24 – Any local evidence for why temperature would be increasing emissions? Because the temperature component seems to be driving the statistical models, it would be useful to understand why there is such a shift around 30oC.

4. Page 10 – line 29 – The explanation of the sharp drop off with temperature and the growing BL height doesn't make very much sense. Can you verify if these changes with time/temperature relate to the simulated PBL height by the model?

5. Page 12, lines 25-30: What are the relative magnitudes of u and v winds in the region? E.g, If the u winds are higher than v (as they frequently are), would that explain the improved correlations? Additionally, how does this relate to the wind parameterization that is implemented? E.g., if the magnitude of v winds are always below the threshold, then this might explain the lack of correlation.

6. Page 13 – line 10: Can you explain why the modeled RH correlation would be higher given that the model simulated observed RH with good fidelity?

7. Page 13 (and conclusions, page 15 line 25)– It is unclear why wind speed is not a good predictor in your models. Can this be compared with other studies to place this finding in context.

8. Section 4.3: There is little discussion of why performance is different at the eight

different sites. This could be expanded in a revised manuscript.

Editorial comments

1. Page 2 – line 1: Does this mean that throughout the world Melbourne has the highest population of allergy sufferers, or just in Australia? Does the reference support this? It appears to discuss the European community.

2. Page 2- lines 30-35: There are a few other dispersion model studies missing (e.g., oak study in CMAQ; Pasken and Pietrowitz, 2005)

3. Page 3 – line 10 – Does the lack of other wind-driven pollination species also apply to all other grasses? It is unclear if there are other Australian species that would be contributing to the grass pollen count outside of the ryegrass. My understanding is that different genuses of the Poaceae family are very hard to distinguish in daily counts.

4. Page 3 – line 30: Genus distinctions can be made based on volumetric sampling methods. Additionally, it would be helpful to note how many different pollen types (e.g, at the family or genus level) are counted at the Australian sites.

5. Page 6 – line 10 – Other studies (e.g., van Hout et al, 2008; Viner et al, 2010) have shown an early morning timing peak. Any ideas as to why ryegrass might be different?

6. Figure 2- It's hard to see the correlation between the EVI and pollen count. A climatological average would show the timing better.

7. Figure 4 – Can you put the V1 and V2 x axes on the same scale to improve visual comparisons?

8. Page 9 – I'm confused about the loss factor in Equation 11 – how were these set, and how are they different from a regular deposition rate?

9. Page 13 – lines 18-20: Move this text to the figure caption.

References

[Figure]

Pasken, R. and J.A. Pietrowiez (2005) Using dispersion and mesoscale meteorological models to forecast pollen concentrations, Atmos. Environ., 39, 2689-7701.

Van Hout et al., (2008) The influence of local meteorological conditions on the circadian rhythm of corn (Zea mays L.) pollen emission, agricultural and forest meteorology, 148, 1078-1092.

Viner, B.J., M.E. Westgate and R.W. Arritt, 2010: A model to predict diurnal pollen shed in maize. Crop Science 50, 235-245.

---

## Referee Comment (RC2) · Anonymous Referee #2 · 23 Apr 2019

**Anonymous Referee:**

**General comments:** I would like to thank Ms Emmerson and colleagues for submitting an interesting and detailed manuscript which describes a thorough assessments of the different prognostic approaches used in the grass pollen emission-source methodologies in their model (VGPEM1.0 ), applicable to Victoria, Australia. The publication also includes an excellent review of most of the relevant material on pollen emission modelling approaches to date, nicely summarising the limitations of regression versus statistical approaches. Whilst focussed on specific pollen type as might be expected and germane to this study region (*Poaceaea* ) this is justified due to the significant health impacts, as highlighted by the thunderstorm induced asthma dispersal incident referenced, and the results produced are useful. The link to EVI is also useful. As discussed in section 3 of their manuscript the modelling approach does not include some of the more detailed mechanisms associated with pollen emission, that may impact pollen production and re-suspension etc. which may influence the variances discussed, but this is justified due to the lack of quantitative observational  as well as modelling information for these mechanisms.  The POD, FAR and ETS approach is well described although some context would be helpful.

Pollen types and sources can of course vary enormously across the globe but this work provides much needed evidence with good statistical analysis for different approaches and their applications to other regions and pollen types.

There will be many in the community who welcome this work as highlighting a growing need for pollen and other allergen/pathogen consideration in pollution modelling.

In my opinion the manuscript is worthy of publication. Only very small changes are needed in order to clarify certain areas (for the non-pollen experts) with answers to minor questions to help support one or two sections with occasional reflection in the conclusions.

**Comments**

1. There a couple of other studies that might be worth referencing to place this work in a more global context, e.g. Lake et al. (2017),and Pasken and Pietrowitz

2. Section 2 Observations and characteristics of grass pollen. You list the pollen risk categories specific to Victoria/Australia. How do  these factors relate to international risk factor league tables e.g. in the USA (where other factors such as Pollen and Overall national Capital Risk Factors for individual cities are produced or clinical risk factors in the EU usually in terms of grains per year, e.g. Agnew et al. 2018, Int J Environ Res Public Health).

3. Section 3.1.6  Statistical Models. The limitation of the statistical models due to coarseness of the temporal training data (daily) is understood, however a sentence might be useful here explaining how this limitation is linked to the actual physical emission mechanism timescales via the gross timing function and day to day expected variation.

4. You show the non-linear relationships between VI, V2 model pollen responses  and temperature, rainfall and relative humidity (Figure 4) – I assume the grayed areas represent the variances in each case? – So, can a brief sentence or two be included in this section to explain/summarise how these meteorological drivers actually physically relate to the pollen emission mechanisms please?

5. How representative are these responses, especially temperature, for Australia generally and for this pollen type in particular? I am thinking of the study by Viner et al, 2010, as also pointed out by the Editor regarding the timing response.

6. Can this statistical approach be robust enough to respond to inter-annual variations?

7. How representative might these relationships shown in Figure 4 be with respect to inter-annual variation (will the EVI approach take this into account)?

8. How do these dependencies, especially with temperature, compare to other pollen types described elsewhere in the literature as this obviously has implications for the risk factor analysis particularly if it is to be extended to other regions? A brief sentence on this might help with context.

9.Page 13, line 6. You state that "Transport of pollen from the productive grasslands in the west of Victoria to Melbourne would rely on the U wind vector being modelled accurately, however the model lifetime of these pollen grains is 6 hours over a height of 1 km; too short for pollen emitted near Hamilton to reach Melbourne.`'

You pose the initial question suggesting you will justify this but then ignore the point by assuming it is modeled correctly in order to justify the conclusion that these grasslands were not the source. Perhaps you could rephrase this sentence to make it less confusing?

10. Page 13, line 8. You state "We extracted the boundary layer height from the model (unavailable in the observations), which showed that the modelled grass pollen is more strongly correlated to atmospheric dilution (average $r=0.61$) than it is to temperature (average $r=0.44$). The model RH is more negatively correlated with grass pollen levels (average $r=-0.52$) than is observed." Now going back now to Page 10, where you state that, "This decline in concentrations may be due to increased boundary layer heights (and thus greater effective dilution) rather than a decrease in emissions."

The latter statement I agree with but is this consistent with your statement about that a growing boundary layer depth is accompanied by a sharp drop in temperature – is this correct? Have I read this correctly? One might expect that the concentrations are inversely related to the volume of air available for vertical mixing from the surface to the boundary layer top, or more precisely the mixed layer. So, increasing boundary layer height due to increasing convection during the day (and surface temperature)~ would lead to increased dilution due to turbulent mixing and dispersion in the lower boundary layer (unless in a zone impacted by strong recirculation from convective outflow or topographic influence). I believe this is generally observed in Melbourne. A drop in temperature response does not seem consistent?

As I understand it the impacts of summer versus winter boundary layer height development can significantly influence pollutant concentrations in Melbourne whilst the wind direction, e.g. from the local hills and vegetation sources in summer, influences the background tracer concentrations (Coutts et al. 2007, Atmos.Env.)? In addition sea-breezes can also be important in Melbourne and I understand the wind rose for Melbourne displays a very strong annual N-S bimodality with higher frequencies of average winds from the N but of course higher frequencies of much stronger winds from the ocean, S (BoM)?

Perhaps a sentence or two describing the known evolution of the boundary layer height with time of day in the measurement period specific to Melbourne would help put this section in context. It would also be useful for the general readers (even if only by reference to previous work. I note you state there were no contemporaneous observations)?

Although it is not necessary to reference this study the issue of significant variation of pollen concentrations with height may need to be discussed here, e.g. see Damialis, et al. (2017), Nature Scientific Reports. The latter compares surface, tower and aircraft measured pollen concentrations with altitude.

11. Perhaps you could include a brief summary of the wind climatology (your U and V components) as this is central to predicting wind pollinated species. This would also be helpful to place your wind thresholds in context, especially in terms of how these contribute to emission mechanisms and the correlation (or lack thereof) you observe with these thresholds.

12. Section 5 Conclusions. How do your conclusions regarding the wind and RH correlations in particular compare with European and US studies?

13. Would it be helpful to include a statement concerning how much variation the smoothed statistical approach potentially might miss over and above the seasonal maxima?

**References**

Damialis, et al. (2017), Estimating the abundance of airborne pollen and fungal spores at variable elevations using an aircraft: how high can they fly? Nature Scientific Reports, 7, Article number: 44535.

**Lake, IR, Jones, NR, Agnew, M, et al. Climate change and future pollen allergy in Europe. *Environ Health Perspect.* 2017; 125: 385- 391.**

**Pasken, R. and J.A. Pietrowiez (2005) Using dispersion and mesoscale meteorological models to forecast pollen concentrations, Atmos. Environ., 39, 2689-7701.**

**Viner, B.J., M.E. Westgate and R.W. Arritt, 2010: A model to predict diurnal pollen shed in maize. Crop Science 50, 235-245.**

---

## Author Comment (AC1) · 13 May 2019

**Reviewer #1**

*General comments: This manuscript describes the implementation of several different emissions parameterizations to prognostically simulate grass pollen in one region of Australia. The paper is relatively narrow in scope in that it is focused on one type of pollen (grass) in one small region of the globe. That said, it will be useful model for the region and has the potential to be applied to other regions for grass pollen. My comments are mostly minor and regarding changes for clarity and additional supporting information. I recommend the article for publication following these revisions.*

We thank reviewer #1 for their helpful comments on our manuscript. This work is the inaugural version of VGPEM, which we hope can be expanded to cover other pollen taxa suited to all of Australia. The choice of ryegrass pollen in this initial version was because of its high human allergenic properties.

Line numbers review to the Discussions version of the manuscript.

**Comments on the analysis**

*1. Regarding the statistical model functions shown in Figure 4: It is helpful to see what the dependencies look like, but it is unclear how these meteorological factors physically relate to what we know about pollen emissions. Can these observed dependencies be related to any physical processes?*

As these statistical relationships are based on the pollen concentrations at the receptor sites (and therefore not the emissions), the relationships take into account transport and dilution effects.

We suggest that increased dilution due to an increased boundary layer height during warmer days might be responsible for the fall in the pollen response with temperature. The pollen response also decreases with increased relative humidity, as humid conditions would prevent pollen release from the plant.

Zink et al (2013) also show nonlinear temperature and RH functions describing pollen emission, which are similar to ours. They show an emission function for temperature peaking at 22°C then declining, and a relationship with relative humidity decreasing emissions between 50% and 90%. These relationships were achieved through minimization of errors between model and birch pollen measurements. Our relationships are similar, with temperature peaking at ~25°C and RH decreasing above 40% in V1 and 20% for V2. Sofiev et al (2013) suggest that pollen emission is neither inhibited nor promoted outside of 50-80% RH.

In terms of rainfall, Sofiev et al (2013) uses 0.5 mm hr$^{-1}$ (the grid cell average rate) is taken as the threshold suppressing the pollen emission. Our rainfall term shows a sharp decline until about 2 mm day$^{-1}$, after which little additional pollen suppression occurs, although there is considerable uncertainty given the infrequent high-rainfall days.

Replace text on page 10 line 27. "The statistical parameterisations were based on ambient pollen concentrations rather than emissions, and thus the non-linear terms take into account transport and dilution processes. The temperature response in both models increased until 25 to 30°C. The decline in pollen response at higher temperatures is likely due to the dilution with a higher planetary boundary layer (associated with higher temperature); in this case, the assumption of declining emissions with increased temperature is likely incorrect. There is relatively little non-linearity with humidity, and the general trend is for increased concentrations (or emissions) in dryer conditions; this is explained by the drying required for anther dehiscence. The rainfall term shows a sharp decline until about 2 mm day$^{-1}$, after which little additional pollen suppression occurs, although there is considerable

uncertainty given the relative paucity of high-rainfall days. The suppression of grass pollen concentrations (or emissions) is likely due to the low potential for anther dehiscence in moist conditions, and wet deposition of ambient pollen."

*2. Risk category definitions: Page 4 – line 14 – Can you place these count categories (low, moderate, high, extreme) in context as compared to other counts in other regions (e.g., Europe, US)? This would give readers an idea of how high/low Australian counts are relative to other locations in the world. Additionally, for the evaluation based on risk categories on page 12, how do uncertainties in these thresholds influence the analysis?*

Table of pollen count categories for grass where possible. P="pollen"

|  | Australia | MeteoSwiss (Zink et al 2013) | UK Met Office (Osborne et al 2017)[a] | US National Allergy Bureau[b] |
|---|---|---|---|---|
| Low | <19 | <10 | <29 | 1<P<4 |
| Medium | 20<P<49 | 10<P<70 | 30<P<49 | 5<P<19 |
| High | 50<P<99 | 70<P<300 | 50<P<149 | 20<P<199 |
| Extreme | >100 | >300 | >150 | >200 |

a Grass pollen. UK Met Office have 4 different grading systems dependent on taxa.
b https://www.aaaai.org/global/nab-pollen-counts/reading-the-charts

Text added to page 4 line 15. "The Australian grass count categories are similar to those used in the UK and Europe at the low and medium count categories, but the Australian extreme category is reached at pollen counts up to 3 times lower than Europe and the US (Zink et al., 2013; Osborne et al., 2017; US National Allergy Bureau)."

And correction at page 4 line 14 "…graded low if the count is 19 m$^{-3}$ or less"

Figure 7 deals with the second point nicely, as modelled pollen concentrations falling just outside the defined risk category are defined as a 'miss', even though the modelled value may be just outside the category bin. In figure 7 we examine the Gerrity score, probability of detection, equitable threat score, and false alarm rate based on whether the model captures the high pollen risk category correctly. The r correlation and root mean squared error are only based on how close the modelled pollen is numerically to the observed pollen, and therefore 'decategorised'.

The categorised Gerrity score gives a worse result if the model misses by more than one category. The model/observed comparison is assessed using a weighted matrix. The further from the diagonal the model is, the weightings decrease (negative) producing a worse score. In this paper we present a range of statistics which assess different aspects of the model skill.

Add text to page 12 line 6. "Statistical evaluations using categorised and decategorised pollen counts will show how the Australian grass pollen thresholds impact our results."

We see that the best performing methods in each of the panels in figure 7 do not change much, with E8, E9 and E10 performing best in all cases. In terms of the uncertainty in the Australian grass pollen thresholds, the difference between the categorised and decategorised skill tests is small, ranging between 0.1 and 0.2 units.

| | | E8 | E9 | E10 |
|---|---|---|---|---|
| Average r | Decat | 0.54 | 0.47 | 0.65 |
| Average RMSE/100 | Decat | 0.55 | 0.51 | 0.45 |
| Average Gerrity | categorised | 0.44 | 0.32 | 0.42 |
| Average (r and RMSE) | | 0.545 | 0.49 | 0.55 |
| Difference (above – Gerrity) | | 0.105 | 0.17 | 0.13 |

Add text to page 14 line 11. "Comparing the results of the decategorised Pearson correlation and RMSE against the categorised Gerrity score yields minor differences between 0.1 and 0.2 units, and suggests the Australian grass pollen thresholds influence the analysis by ~15%."

*3. Page 6 line 24 – Any local evidence for why temperature would be increasing emissions? Because the temperature component seems to be driving the statistical models, it would be useful to understand why there is such a shift around 30oC.*

Page 6 line 24 refers to the immediate timing function which is used for pollen emission methodologies and not the two statistical models. The temperature function applies a sliding scale between 6°C and 24°C, the function becoming 0.95 at temperatures higher than 24°C. These equations are not related to those from the statistical model on page 10.

Text changes concerning the pollen emission response to temperature above 30 °C are included in our answers to comment 1.

*4. Page 10 – line 29 – The explanation of the sharp drop off with temperature and the growing BL height doesn't make very much sense. Can you verify if these changes with time/temperature relate to the simulated PBL height by the model?*

We are not saying the boundary layer increases with dropping temperatures, rather higher temperatures increase the boundary layer and cause more dilution. Thus the pollen concentrations decrease.

[Figure]

There is a large increase in model boundary layer with higher temperatures, particularly above 30°C.

The average modelled boundary layer on November days below 25°C = 282 m, max =1742 m, and on days above 25°C =378 m, max=3083 m.

Add sentence on page 10 line 29. "On days in November where the temperature is above 25°C, the maximum modelled boundary layer height is nearly double the height modelled on days below 25°C."

*5. Page 12, lines 25-30: What are the relative magnitudes of u and v winds in the region? E.g, If the u winds are higher than v (as they frequently are), would that explain the improved correlations? Additionally, how does this relate to the wind parameterization that is implemented? E.g., if the magnitude of v winds are always below the threshold, then this might explain the lack of correlation.*

The observed (and modelled) average and maximum V winds are higher than the U winds for all the sites in west and central Victoria. The Max U winds are higher in the east of Victoria (Shepparton and Latrobe Valley, valley runs E-W so not surprising), but the difference is not as large as in the west.

| AWS site | Average U obs (model) | Max U obs (model) | Average V obs (model) | Max V obs (model) |
|---|---|---|---|---|
| Hamilton airport | -0.07 (0.17) | 8.7 (6.58) | -0.82 (-0.54) | 13.53 (10.04) |
| Ballarat aerodrome | 0.46 (0.10) | 9.68 (6.20) | -0.86 (-0.55) | 13.9 (10.13) |
| Bendigo airport | 0.65 (0.23) | 8.05 (4.61) | -1.35 (-1.05) | 10.14 (7.36) |
| Geelong racecourse | -0.56 (-0.30) | 5.83 (5.59) | -1.21 (-0.87) | 10.15 (7.76) |
| Melbourne Olympic park | -0.24 (-0.43) | 4.75 (5.50) | -0.63 (-0.73) | 9.3 (8.51) |
| Scoresby | -0.06 (0.01) | 4.75 (4.97) | -0.35 (-0.38) | 15.48 (8.70) |
| Shepparton airport | -0.18 (0.01) | 9.21 (6.51) | -0.92 (-0.74) | 8.49 (6.20) |
| Latrobe valley airport | -1.00 (-0.12) | 7.24 (5.29) | -0.35 (-0.12) | 6.7 (5.29) |

The emission parameterisation uses wind speed rather than the wind components U and V independently, therefore there are no U and V thresholds. There is no threshold minimum wind speed either, and the function yields 0.33 at a wind speed of 0 m/s. We do use a saturation wind speed of 5 m s$^{-1}$ to scale the model wind speeds, which Sofiev et al (2013) describes as the maximum speed that wind actively helps the release of pollen.

Add text to page 7 line 1 to minimise confusion about there not being a lower wind threshold: " …lower rate (0.33 for fstagnant) in still conditions…"

We add text to page 7 line 4: "… above which the wind speed does not promote the release of pollen."

Neither the above analysis of U and V nor this saturation wind speed explains the lack of correlation. Individual correlations for observed data are not strong (0 and 0.4) for U and V. average U correlation was 0.32 and average V is 0.22, which isn't a huge improvement.

I think the better U correlation is more a case of the geography of the pollen source regions being west of most of the pollen count sites, and thus the east-west wind component has a stronger relationship than north-south. We have also included wind roses for each AWS site in the supplementary material, together with a brief wind climatology of the 2017 season on page 12.

Include text on page 12 line 29. "We include a wind rose for each AWS site in the supplementary section to determine the strength of the winds. The roses show a strong southerly influence, corresponding with the afternoon sea breeze at most sites apart from Churchill, located within an east-west aligned valley. Sites further west in Victoria (Hamilton and Creswick) also show a northerly influence, generally with a greater percentage of wind speeds above 4 m s$^{-1}$ than elsewhere."

*6. Page 13 – line 10: Can you explain why the modeled RH correlation would be higher given that the model simulated observed RH with good fidelity?*

Here we have difficulty in that the observations of pollen are not coincident with measured meteorological variables. However we are able to compare the observed meteorology with the model as we can extract model variables from any location in the grid. Thus the comparison of modelled RH with observed RH was at the same location. The comparison of observed RH at the AWS with observed pollen at the count sites was less strong, as the pollen count sites are some 10 – 29 km distant from the AWS sites.

The model is a simplification of reality, and modelled pollen has been coded to have a strong dependence on RH. Thus it is not surprising that the modelled correlations are stronger than those observed.

Added text to page 13 line 11. "The observed relationship may be weaker as the pollen measurements are not coincident with the AWS."

*7. Page 13 (and conclusions, page 15 line 25)– It is unclear why wind speed is not a good predictor in your models. Can this be compared with other studies to place this finding in context.*

Wind speed used alone in the intermediate timing is not a good predictor of pollen. Sofiev et al (2013) suggests wind promotes the pollen emission, but is not solely responsible. A plant needs to flower first before pollen is released, which tends to be controlled by temperature. After flowering, pollen that is ready to be released is easily picked up by the wind. However once this supply is exhausted, the strength of the wind does not matter (the saturation wind speed is 5 m s$^{-1}$). Sofiev et al (2013) also sees a low correlation with wind speed, suggesting that stronger wind speeds increase the emission rate but also increases ventilation and turbulent mixing.

Zink et al (2013) add "Theoretically, if unlimited amounts of pollen were available, higher wind speeds would yield stronger entrainment, and hence more airborne pollen. In reality, this is limited by the fact that at a certain point, the flowers will run out of pollen grains."

Viner et al (2010): "An unexpected result was the absence of a relationship between wind speed and pollen shed in our observations. However, we measured the same range of pollen shed rates at 1 m s–1, the lowest reliable measurement of wind speed by our anemometer, and 5 m s–1, indicating that only a light wind is necessary for pollen shed and stronger winds may not necessarily cause more pollen to be shed."

Add text to page 13 line 22. "…which provided poor prediction skill scores (average r=0.25, 0.18 and 0.17 respectively), similar to results by Viner et al (2010) and Zink et al. (2013). Wind promotes pollen emissions, but the plant must flower first - a process not controlled by wind speed (Sofiev et al., 2013)."

*8. Section 4.3: There is little discussion of why performance is different at the eight different sites. This could be expanded in a revised manuscript.*

The results section discusses how model performance tends to be better at sites outside of the city. In order for pollen to get to city locations there is more transport involved, and the pollen size is large and dense. Perhaps local sources are missing. There is also increased heat and turbulence in the city, and being relatively close to the coast introduces sea breezes. The strength and frequency of these southerly winds are shown in the supplementary for each site.

Most of the pollen emission methodologies rely on the distribution of pasture grass. We include satellite maps of each site and their surrounding fraction of pasture grass cover in the supplementary section. As the pollen has a relatively short lifetime, those sites located next to grass pixels tend to be modelled better than those which are not. The least well modelled site is Geelong, which has very strong southern ocean influences (approx. 15 km from the coast. There are limited grass pixels between the coast and the pollen count site.

Include sentence on page 8 line 33. "We include larger scale maps of the pasture grass coverage surrounding the pollen count sites in the supplementary material."

Add text to page 14 line 11. "The sites vary considerably in terms of surrounding land use, whereas all the pollen in the model comes from pasture grass. This impacts the individual site performance against the pollen observations. Hamilton, Dookie and Churchill are close to pollen source areas. Creswick is surrounded by forest. The Burwood and UoM sites are in heavily built up areas with green space, which is not included in the model pasture grass maps."

Include text on page 14 line 10. "The wind rose for Geelong shows the strong Southern Ocean influence, and there are few grass filled pixels between the coast and pollen count site which the model relies upon (supplementary)."

The model assumption is that all grass pollen comes from pasture grass. The meteorological parameters are all modelled very well, therefore individual site performance could come down to whether we have the correct spatial distribution of the pollen source regions. Inverse modelling could highlight discrepancies between our pasture grass emission source areas, and other grass land use categories contributing to grass pollen. However this is out of scope of the current paper.

Include text at page 15 line 18. "Inverse modelling could highlight where other grass land use categories contribute to grass pollen."

To test whether the pollen observations at all the count sites are related, we plot $r^2$ correlations. Observations in yellow, left, and modelled E10 scenario right.

|      | HAM  | CWK  | BGO  | GEE  | UOM  | BUR  | DOK  | CHU  |
|------|------|------|------|------|------|------|------|------|
| HAM  |      | 0.78 | 0.91 | 0.79 | 0.86 | 0.79 | 0.85 | 0.59 |
| CWK  | 0.55 |      | 0.84 | 0.70 | 0.80 | 0.85 | 0.68 | 0.83 |
| BGO  | 0.10 | 0.17 |      | 0.89 | 0.90 | 0.88 | 0.91 | 0.69 |
| GEE  | 0.25 | 0.37 | 0.19 |      | 0.92 | 0.88 | 0.80 | 0.70 |
| UOM  | 0.37 | 0.55 | 0.31 | 0.38 |      | 0.96 | 0.79 | 0.75 |
| BUR  | 0.28 | 0.40 | 0.22 | 0.36 | 0.72 |      | 0.74 | 0.84 |
| DOK  | 0.00 | 0.06 | 0.39 | 0.05 | 0.05 | 0.05 |      | 0.57 |
| CHU  | 0.44 | 0.56 | 0.06 | 0.17 | 0.37 | 0.34 | 0.02 |      |

Add text to page 15 line 16."Correlations between observed pollen at each site are not particularly strong (average $r^2$ = 0.28), suggesting that the pollen sources may not be related, or are highly localised. The modelled correlations between all sites are very strong because they share the same pollen source characteristics (average $r^2$=0.80)".

**Editorial comments**

*1. Page 2 – line 1: Does this mean that throughout the world Melbourne has the highest population of allergy sufferers, or just in Australia? Does the reference support this? It appears to discuss the European community.*

Table 1 (top line) of Bousquet et al (2008) shows Melbourne has the highest allergy rate in the world. Melbourne has the highest prevalence of nasal allergy (46), and the highest atopic nasal allergy (32) of any region studied. The title of the Bousquet paper is somewhat misleading as it refers to the European Community Respiratory Health Survey.

Alter text at page 2 line 1 to read "Melbourne, on the south east coast of Australia, has the highest prevalence of allergic rhinitis in the world (Bousquets et al 2008). Melbourne, in the State of Victoria, is a city of approximately 4.9 million inhabitants."

*2. Page 2- lines 30-35: There are a few other dispersion model studies missing (e.g., oak study in CMAQ; Pasken and Pietrowitz, 2005)*

Add text to page 2 line 32. "An understanding of pollen release biology together with accurate meteorological data is crucial for pollen forecasting (Pasken and Pietrowitz, 2005)."

Add text to page 2 line 33. "Indeed, climate induced spread of ragweed is predicted to double the number of Europeans suffering allergic responses by 2060 (Lake et al, 2017)."

*3. Page 3 – line 10 – Does the lack of other wind-driven pollination species also apply to all other grasses? It is unclear if there are other Australian species that would be contributing to the grass pollen count outside of the ryegrass. My understanding is that different genuses of the Poaceae family are very hard to distinguish in daily counts.*

Correct, grass pollen is counted, not ryegrass pollen. We use the physical properties of ryegrass to represent all grass pollen in the model, as it is the major allergen.

Other Melbourne researchers also comment on the dominance of ryegrass. Ong et al (1995) suggest that the Melbourne count, which is the most resolved for species, that it is all introduced species such as ryegrass and canary grass. The Medek et al (2016) paper talks about C3 grasses dominating in the temperate southern region of Australia (e.g. Victoria), mainly pasture grasses. And Schappi et al (1998) "In the cool temperate climate of this region, rye-grass (Lolium perenne) pollen is more abundant than pollen from other grass species (Smart and Knox, 1979), since rye-grass is grown extensively as pasture grass in this area and has a very high pollen output (Smart et al., 1979)"

Seminal work done by Smart et al (1979) counted anthers per flowering spike of grasses to the north of Melbourne. "Some grasses, for example brome and wild oat, have a low pollen output while others, particularly agricultural grasses, for example ryegrass, Yorkshire fog and canary grass, have a very high

output. The high pollen producers are all introduced cool temperate pasture grasses. Native grasses, for example wallaby and kangaroo grass, are low pollen producers." Combined with the large areas of land given over to ryegrass production, ryegrass must be the dominant pollen producer in Victoria.

Alter text on page 3 line 10 "Native Australian grasses such as wallaby and kangaroo grass are generally not wind-pollinated and produce little pollen, whilst introduced agricultural pasture grasses such as ryegrass (Lolium perenne) and canary grass are high pollen emitters (Smart et al., 1979). Ryegrass is grown extensively in Victoria."

Change text at page 3 line 13 "attributed to ryegrass"

Add text to page 4 line 3. "DNA sequencing at UoM indicates ryegrass can contribute 60 to 90% of grass pollen counted over the 2016 pollen season (personal communication, E. Newbigin)."

*4. Page 3 – line 30: Genus distinctions can be made based on volumetric sampling methods. Additionally, it would be helpful to note how many different pollen types (e.g, at the family or genus level) are counted at the Australian sites.*

Grass pollen cannot be further categorized by microscopy (not to genus or species).

20 pollen taxa are listed in Haberle et al (2014) as species contributing more than 80% of annual pollen in Australia and New Zealand, of which 15 are found in Melbourne.

Add text page 4 line 1 "In Victoria, routine pollen counting since 2017 distinguishes between 15 pollen taxa, with Haberle et al. (2014) finding 70 % of the total pollen…, however we concentrate on….."

*5. Page 6 – line 10 – Other studies (e.g., van Hout et al, 2008; Viner et al, 2010) have shown an early morning timing peak. Any ideas as to why ryegrass might be different?*

The van Hout and Viner references refer to corn. I think the key here is that ryegrass and corn are entirely different plants. Ryegrass flowers in spring whereas corn flowers in summer. The time of pollen release also depends when pollen can get into the air (ie needs vertical uplift). If cool and damp (spring), the time of pollen release is likely to be later in the day.

Melbourne evidence for the later peak comes from Smart & Knox (1979) who measured rye-grass emissions with two daily peaks [early morning and late afternoon], the latter afternoon peak being 3.5 times higher than the morning. We have approximated this relationship by centring the release at noon, with a 4 hour standard deviation.

[Figure]

Pollen emission timing figure from Smart and Knox (1979)

Plants differ as to what conditions they need to dehisce: Zink et al (2013) "some grasses need high relative humidities for the opening of their anthers since they have to swell in order to crack." In Viner et al (2010) "Once an anther has lost sufficient moisture, the structure opens at its tip, releasing the mature pollen grains within."

Alcazar et al (2019) "Different grass species flower at different times of the year (Beddows 1931; Jones 1952) and day; this may affect the diurnal patterns in pollen from this family. As an example, Agrostis and Festuca flower at midday, whereas Anthoxanthum and Holcus flower in the morning or late afternoon (Hyde and Williams 1945; Peel et al. 2014)."

Peel et al (2014) "Vapour Pressure Deficit may be considered a proxy for the drying power of the air, and greater VPD earlier in the day may thus lead to earlier drying, emission and concentration peaks."

However I'm not sure any of the above is relevant to our study. Alter/add text to page 6 line 10. "…(measured as the number of exposed anthers) occurs in the early afternoon. As ryegrass flowers in spring when mornings are cool and damp, the anthers need to dry before pollen is released."

*6. Figure 2- It's hard to see the correlation between the EVI and pollen count. A climatological average would show the timing better.*

We present climatologies of the time series for figure 2 panels a and b.

Change figure 2 caption to read "(a) 16 year climatology in EVI from South West Victoria…."

*7. Figure 4 – Can you put the V1 and V2 x axes on the same scale to improve visual comparisons?*

Done.

*8. Page 9 – I'm confused about the loss factor in Equation 11 – how were these set, and how are they different from a regular deposition rate?*

Zink et al (2013) also propose a production and loss model, where the pollen reservoir is considered to be pollen available for release but has not left the plant. The pollen can rest on leaves etc. A loss can be incurred at this stage by animals brushing past or loss to the ground. Zink then suggest this 'random' loss process is similar to a half-life, such that after 12 hours, half the reservoir is lost. We extend this idea to provide a variable loss rate which is accelerated in wet conditions.

Rewrite page 9 line 7. "…and once exhausted by in-plant dry and wet deposition, or pollen release, the pollen reservoir is only replenished at a finite rate. Scenario E7 is a production-loss model for this pollen reservoir."

Rewrite section beneath equations on page 9. "where emissions, E are set to be the product of the available pollen reservoir, A and the instantaneous emission factor, I at grid-point (x,y) at time t. $\delta t$ is the model time-step. The pollen produced, P is given by the product of the spatial and gross-timing terms, proportional to the fraction of the grass pollen season covered between t and t + $\delta t$, L is the amount lost between t and t + $\delta t$, T is the total length of the grass pollen season and $\lambda$ is the loss rate due to direct deposition before the pollen leaves the plant. This loss can occur direct to the ground or via animals brushing past, and differs from the in-atmosphere wet and dry deposition rates. Zink et al

(2013) suggest this loss process is similar to a half-life, which we extend to provide a variable loss rate accelerated in wet conditions. The loss decay parameter ($\lambda$), is defined as a piece-wise polynomial function based on the rain rate such that pollen has a half-life on the plant of two days in dry conditions and 12 hours in wet conditions, with the latter corresponding to a rain rate of 2 mm $h^{-1}$ .

*9. Page 13 – lines 18-20: Move this text to the figure caption.*

Already in figure caption so have removed from the main text.

**References**

Alcázar, P., Ørby, P.V., Oteros, J. et al. Cluster analysis of variations in the diurnal pattern of grass pollen concentrations in Northern Europe (Copenhagen) and Southern Europe (Cordoba). Aerobiologia (2019). https://doi.org/10.1007/s10453-019-09558-2

Bousquet, P. J., Leynaert, B., Neukirch, F., Sunyer, J., Janson, C. M., Anto, J., Jarvis, D., and Burney, P.: Geographical distribution of atopic rhinitis in the European Community Respiratory Health Survey I, Allergy, 63, https://doi.org/10.1111/j.1398-9995.2008.01824.x, 2008.

Erbas, B., Chang, J-H., Newbigin, E., Dhamarge, S. (2007) Modelling atmospheric concentrations of grass pollen using meteorological variables in Melbourne, Australia, International Journal of Environmental Health Research, 17:5, 361-368, DOI: 10.1080/09603120701628693

Haberle, S., Bowman, D., Newnham, R., Johnston, F., Beggs, P., Buters, J., Campbell, B., Erbas, B., Godwin, I., Green, B., Huete, A., Jaggard, A., Medek, D., Murray, F., Newbigin, E., Thibaudon, M., Vicendese, D., Williamson, G., and Davies, J.: The macroecology of airborne pollen in Australian and New Zealand urban areas, PLoS ONE, 9, e97 925, https://doi.org/10.1371/journal.pone.0097925, 2014.

Lake, IR, Jones, NR, Agnew, M, et al. Climate change and future pollen allergy in Europe. Environ Health Perspect. 2017; 125: 385- 391.

Medek, D. E., Beggs, P. J., Erbas, B., Jaggard, A. K., Campbell, B. C., Vicendese, D., Johnston, F. H., Godwin, I., Huete, A. R., Green, 30 B. J., Burton, P. K., Bowman, D. M. J. S., Newnham, R. M., Katelaris, C. H., Haberle, S. G., Newbigin, E., and Davies, J. M.: Regional and seasonal variation in airborne grass pollen levels between cities of Australia and New Zealand, Aerobiologia, 32, 289–302, https://doi.org/10.1007/s10453-015-9399-x, 2016.

National Allergy Bureau. https://www.aaaai.org/global/nab-pollen-counts/reading-the-charts. American Academy of Allergy, Asthma and Immunology.

Osborne, N.J., Alcock, I., Wheeler, B.W. et al. Pollen exposure and hospitalization due to asthma exacerbations: daily time series in a European city. Int J Biometeorol (2017) 61: 1837. https://doi.org/10.1007/s00484-017-1369-2

Pasken, R. and J.A. Pietrowiez (2005) Using dispersion and mesoscale meteorological models to forecast pollen concentrations, Atmos. Environ., 39, 2689-7701.

Peel, R. G., Ørby, P. V., Skjøth, C. A., Kennedy, R., Schlünssen, V., Smith, M., Sommer, J., and Hertel, O.: Seasonal variation in diurnal atmospheric grass pollen concentration profiles, Biogeosciences, 11, 821-832, https://doi.org/10.5194/bg-11-821-2014, 2014.

Smart, I. J. and Knox, R. B.: Aerobiology of Grass-Pollen in the City Atmosphere of Melbourne - Quantitative-Analysis of Seasonal and Diurnal Changes, Aust J Bot, 27, 317–331, https://doi.org/10.1071/Bt9790317, 1979.

Smart, I. J., Tuddenham, W. G., and Knox, R. B.: Aerobiology of Grass-Pollen in the City Atmosphere of Melbourne - Effects of Weather Parameters and Pollen Sources, Aust J Bot, 27, 333–342, https://doi.org/10.1071/Bt9790333, 1979.

Sofiev, M., Siljamo, P., Ranta, H., Linkosalo, T., Jaeger, S., Rasmussen, A., Rantio-Lehtimäki, A., Severova, E., and Kukkonen, J.: A nu35 merical model of birch pollen emission and dispersion in the atmosphere. Description of the emission module, International Journal of Biometeorology, 57, 45–58, https://doi.org/10.1007/s00484-012-0532-z, 2013.

Van Hout et al., (2008) The influence of local meteorological conditions on the circadian rhythm of corn (Zea mays L.) pollen emission, agricultural and forest meteorology, 148, 1078-1092.

Viner, B.J., M.E. Westgate and R.W. Arritt, 2010: A model to predict diurnal pollen shed in maize. Crop Science 50, 235-245.

Zink, K., Pauling, A., Rotach, M. W., Vogel, H., Kaufmann, P., and Clot, B.: EMPOL 1.0: A new parameterization of pollen emission in numerical weather prediction models, Geosci Model Dev, 6, 1961–1975, https://doi.org/10.5194/gmd-6-1961-2013, 2013.

---

## Author Comment (AC2) · 13 May 2019

*General comments: I would like to thank Ms Emmerson and colleagues for submitting an interesting and detailed manuscript which describes a thorough assessments of the different prognostic approaches used in the grass pollen emission-source methodologies in their model (VGPEM1.0 ), applicable to Victoria, Australia. The publication also includes an excellent review of most of the relevant material on pollen emission modelling approaches to date, nicely summarising the limitations of regression versus statistical approaches. Whilst focussed on specific pollen type as might be expected and germane to this study region (Poaceaea ) this is justified due to the significant health impacts, as highlighted by the thunderstorm induced asthma dispersal incident referenced, and the results produced are useful. The link to EVI is also useful. As discussed in section 3 of their manuscript the modelling approach does not include some of the more detailed mechanisms associated with pollen emission, that may impact pollen production and re-suspension etc. which may influence the variances discussed, but this is justified due to the lack of quantitative observational as well as modelling information for these mechanisms. The POD, FAR and ETS approach is well described although some context would be helpful.*

*Pollen types and sources can of course vary enormously across the globe but this work provides much needed evidence with good statistical analysis for different approaches and their applications to other regions and pollen types.*

*There will be many in the community who welcome this work as highlighting a growing need for pollen and other allergen/pathogen consideration in pollution modelling. In my opinion the manuscript is worthy of publication. Only very small changes are needed in order to clarify certain areas (for the non-pollen experts) with answers to minor questions to help support one or two sections with occasional reflection in the conclusions.*

We thank reviewer #2 for these helpful suggestions to improve the manuscript and make it clearer.

**Comments**

*1. There a couple of other studies that might be worth referencing to place this work in a more global context, e.g. Lake et al. (2017),and Pasken and Pietrowitz.*

Lake et al (2017) estimated the number of Europeans suffering allergic responses to ragweed by 2060 will more than double, due to climate change and the estimated spread of the plant. Climate change could prolong the pollen season, changing the pattern of exposure.

Pasken and Pietrowitz (2005) make short-term forecasts of oak pollen in Missouri, USA, finding that the accuracy of the meteorological data was crucial in tandem with an understanding of pollen release biology.

Add text to page 2 line 32. "An understanding of pollen release biology together with accurate meteorological data is crucial for pollen forecasting (Pasken and Pietrowitz, 2005)."

Add text to page 2 line 33. "Indeed, climate induced spread of ragweed is predicted to double the number of Europeans suffering allergic responses by 2060 (Lake et al, 2017)."

*2. Section 2 Observations and characteristics of grass pollen. You list the pollen risk categories specific to Victoria/Australia. How do these factors relate to international risk factor league tables e.g. in the USA (where other factors such as Pollen and Overall national Capital Risk Factors for individual cities are produced or clinical risk factors in the EU usually in terms of grains per year, e.g. Agnew et al. 2018, Int J Environ Res Public Health).*

This comment relates to analysis comment 2 reviewer #1 posed on international pollen classifications. In terms of pollen forecasting having a daily risk classification is crucial. Agnew et al (2018) look at allergic sensitization and disease in children, and relate the impacts to the number of ragweed pollen grains counted per year. Children living in high pollen areas (>5000 grains $m^{-3}$ $year^{-1}$) were more at risk than those living in low pollen regions (<400 grains $m^{-3}$ $year^{-1}$), though the study found that children brought up in rural areas had more resistance than city dwellers.

The Asthma Capital Risk Factors report lists cities in terms of eight contributing factors, pollen being one of them. It is not just about pollen (https://www.aafa.org/media/2119/aafa-2018-asthma-capitals-report.pdf).

Text added to page 4 line 15. "The Australian grass count categories are similar to those used in the UK and Europe at the low and medium count categories, but the Australian extreme category is reached at pollen counts up to 3 times lower than Europe and the US (Zink et al., 2013; Osborne et al., 2017; US National Allergy Bureau)."

Add text to page 4 line 15. "Whilst epidemiological studies commonly use annual pollen totals, we use a daily pollen risk classification system because we aim to predict daily pollen concentrations."

*3. Section 3.1.6 Statistical Models. The limitation of the statistical models due to coarseness of the temporal training data (daily) is understood, however a sentence might be useful here explaining how this limitation is linked to the actual physical emission mechanism timescales via the gross timing function and day to day expected variation.*

The statistical models are trained on daily data, and predict daily pollen concentrations when used at the BoM. However the C-CTM requires emissions at an hourly frequency and uses EVI data and hourly meteorological data to drive the emissions, therefore linking emissions to the day to day expected variation. The gross-timing function is meant to smooth out much of the day-to-day variation, and is modulated by the immediate-timing term when estimating variation in time in the emissions module.

Add text to page 9 line 27. "…thus cannot resolve higher-resolution temporal variation. The gross-timing function smooths out much of the day-to-day variation, and is modulated by the immediate-timing term when estimating temporal variability in the emissions module."

*4. You show the non-linear relationships between VI, V2 model pollen responses and temperature, rainfall and relative humidity (Figure 4) – I assume the grayed areas represent the variances in each case? – So, can a brief sentence or two be included in this section to explain/summarise how these meteorological drivers actually physically relate to the pollen emission mechanisms please?*

Add text to figure 4 caption. "The shaded regions correspond to ± twice the standard error of the GAM term."

Add text to page 10 line 25. "The shaded regions correspond to ± twice the standard error of the GAM term and are greater in regions of the distribution with fewer observations. For example, there were far fewer observations at the upper tail of the temperature range considered, and the standard errors are correspondingly larger."

The second part of the question relates to analysis comment 1 from reviewer #1.

Replace text on page 10 line 27. "The statistical parameterisations were based on ambient pollen concentrations rather than emissions, and thus the non-linear terms take into account transport and dilution processes. The temperature response in both models increased until 25 to 30 °C. The decline in pollen response at higher temperatures is likely due to the dilution with a higher planetary boundary layer. Thus the assumption of declining emissions with increased temperature is likely incorrect. There is relatively little non-linearity with humidity. The general trend is for increased concentrations (or emissions) in drier conditions, explained by the drying required before anther dehiscence. The rainfall response shows a sharp decline until about 2 mm day$^{-1}$, after which little additional pollen suppression occurs, although there is considerable uncertainty given the relative paucity of high-rainfall days. The suppression of grass pollen concentrations (or emissions) is likely due to the low potential for anther dehiscence in moist conditions, and wet deposition of ambient pollen."

*5. How representative are these responses, especially temperature, for Australia generally and for this pollen type in particular? I am thinking of the study by Viner et al, 2010, as also pointed out by the Editor regarding the timing response.*

Relates to editorial comment 5 of reviewer #1. Earlier Melbourne grass pollen GAM modelling by Erbas et al (2007) also found nonlinear terms for temperature, rainfall and RH.

Add text to page 10 line 29. "The shapes of these relationships are similar to those described by Erbas et al (2007) for grass pollen in Melbourne, and also by Zink et al (2013) for birch pollen in Europe."

*6. Can this statistical approach be robust enough to respond to inter-annual variations?*

The statistical approach accounts for inter-annual variation via the EVI time-series at each grid-cell. Higher winter-time peak EVI values are associated with higher cumulative grass pollen counts over the following season. The EVI approach factors in the recent growing conditions (accounting for both temperature and rainfall); this approach appears to be reasonably effective for grass pollen in the Mediterranean style climate of SE Australia.

Add text to page 11 line 10. "The statistical approach accounts for inter-annual variation via the EVI time-series at each grid-cell. Higher winter-time peak EVI values are associated with higher cumulative grass pollen counts over the following season."

*7. How representative might these relationships shown in Figure 4 be with respect to interannual variation (will the EVI approach take this into account)?*

We think the answer to this question relates to that given for the question 6.

The site at Melbourne is the longest running count site in Australia. Pollen data does extend back to the start of the 1990s but there are difficulties finding consistent satellite data at these earlier years to drive that component of the model.

We can see in figure 4 that the relationships for temperature and RH between V1 and V2 are similar. V2 is trained on one extra year at Melbourne and includes data from 7 additional sites than V1, yet the relationships do not change much – i.e., the peak in the temperature response for V1 and V2 are both ~0.5 and peak around 25-30°C, and there is a negative relationship with RH.

*8. How do these dependencies, especially with temperature, compare to other pollen types described elsewhere in the literature as this obviously has implications for the risk factor analysis particularly if it is to be extended to other regions? A brief sentence on this might help with context.*

At this stage we only propose to include grass pollen in an Australia wide model. It is expected that additional training data from other regions will be included at this stage. There are long time series of pollen counts in Sydney, Canberra, Brisbane and from sites in Tasmania. We know from these data that different seasonal timings will be introduced, as tropical grasses emit their pollen at different times of year (Beggs et al 2015).

Add text to conclusions, page 16 line 12 "Additional training data would be included to model pollen in other Australian regions, to account for the different seasonal flowering times of other grass species (e.g. C4 grasses) (Beggs et al. 2015)."

*9. Page 13, line 6. You state that "Transport of pollen from the productive grasslands in the west of Victoria to Melbourne would rely on the U wind vector being modelled accurately, however the model lifetime of these pollen grains is 6 hours over a height of 1 km; too short for pollen emitted near Hamilton to reach Melbourne."' You pose the initial question suggesting you will justify this but then ignore the point by assuming it is modeled correctly in order to justify the conclusion that these grasslands were not the source. Perhaps you could rephrase this sentence to make it less confusing?*

U and V are modelled well when compared to AWS data (fig 5). Investigation of the pollen transport distances are relatively short, due to pollen size and density. Generally the pollen counted locally is predominantly determined by local grass coverage.

Add text to page 13 line 6 instead of existing sentence. "The observed U and V correlations are not strong however, and do not point to particular locations being strong pollen sources. Inverse modelling may help pinpoint productive grass pollen regions for each site."

*10. Page 13, line 8. You state "We extracted the boundary layer height from the model (unavailable in the observations), which showed that the modelled grass pollen is more strongly correlated to atmospheric dilution (average r=0.61) than it is to temperature (average r=0.44). The model RH is more negatively correlated with grass pollen levels (average r=- 0.52) than is observed." Now going back now to Page 10, where you state that, "This decline in concentrations may be due to increased boundary layer heights (and thus greater effective dilution) rather than a decrease in emissions."*

*The latter statement I agree with but is this consistent with your statement about that a growing boundary layer depth is accompanied by a sharp drop in temperature – is this correct? Have I read this*

*correctly? One might expect that the concentrations are inversely related to the volume of air available for vertical mixing from the surface to the boundary layer top, or more precisely the mixed layer. So, increasing boundary layer height due to increasing convection during the day (and surface temperature)~ would lead to increased dilution due to turbulent mixing and dispersion in the lower boundary layer (unless in a zone impacted by strong recirculation from convective outflow or topographic influence). I believe this is generally observed in Melbourne. A drop in temperature response does not seem consistent?*

The statement on page 10 talks about non-linear pollen emissions with temperature, and that pollen emissions decrease at temperatures in excess of 30°C (not the temperatures themselves decreasing). We think this drop in pollen emissions at high temperature is because of an increase in boundary layer height causing dilution. We adjust the sentence on page 10 line 28 to make this point clearer:

"The temperature response in both models increased until 25 to 30°C. The decline in pollen concentration at higher temperatures is likely due to the dilution with a higher planetary boundary layer (associated with higher temperature)."

*As I understand it the impacts of summer versus winter boundary layer height development can significantly influence pollutant concentrations in Melbourne whilst the wind direction, e.g. from the local hills and vegetation sources in summer, influences the background tracer concentrations (Coutts et al. 2007, Atmos.Env.)? In addition sea-breezes can also be important in Melbourne and I understand the wind rose for Melbourne displays a very strong annual N-S bimodality with higher frequencies of average winds from the N but of course higher frequencies of much stronger winds from the ocean, S (BoM)?*

*Perhaps a sentence or two describing the known evolution of the boundary layer height with time of day in the measurement period specific to Melbourne would help put this section in context. It would also be useful for the general readers (even if only by reference to previous work. I note you state there were no contemporaneous observations)?*

[Figure]

There are no coincident observations of boundary layer height, and it is not routinely measured by the BoM. The Coutts et al (2007) paper only describes CO2 fluxes and not how the Melbourne boundary

layer evolves. I have extracted the boundary layer height variable from the ACCESS model and plotted a diurnal average across November 2017.

Add text to page 13 line 10. "Average modelled diurnal boundary layer evolution during November 2017 in Melbourne is to increase after sunrise at 05:00 (AEST) to a peak of 1780 m at 13:00. The height begins to decline during the afternoon coincident with a southerly seabreeze, but is still above 1200 m at 17:00. The nocturnal boundary layer is around 200 m."

*Although it is not necessary to reference this study the issue of significant variation of pollen concentrations with height may need to be discussed here, e.g. see Damialis, et al. (2017), Nature Scientific Reports. The latter compares surface, tower and aircraft measured pollen concentrations with altitude.*

Given what we know about the short model lifetime of pollen, we see 1-2 orders of magnitude decrease in pollen concentrations between the surface and 250 m. Damialis et al (2017) found most of their grass pollen at the ground level rather than aloft, corroborating our finding.

Add text where the boundary layer is discussed, "Over 77% of grass pollen is found at ground level (Damialis et al 2017) due to its size and density. The lifetime of our model pollen over 1 km is 6 hours."

*11. Perhaps you could include a brief summary of the wind climatology (your U and V components) as this is central to predicting wind pollinated species. This would also be helpful to place your wind thresholds in context, especially in terms of how these contribute to emission mechanisms and the correlation (or lack thereof`) you observe with these thresholds.*

Include text on page 12 line 29. "We include a wind rose for each AWS site in the supplementary section to determine the strength of the U winds. The roses show a strong southerly influence, corresponding with the afternoon sea breeze at most sites apart from Churchill, located within an east-west aligned valley. Sites further west in Victoria (Hamilton and Creswick) also show a strong northerly influence, generally with a greater percentage of wind speeds above 4 m s$^{-1}$ than elsewhere."

*12. Section 5 Conclusions. How do your conclusions regarding the wind and RH correlations in particular compare with European and US studies?*

Sofiev et al (2013, fig 4) also show observed pollen being negatively correlated with RH, and has very little correlation with observed wind speed. The reason for the low correlation with wind speed is "the result of two competitive effects. The stronger wind speed increases the emission rate but also improves ventilation and promotes turbulent mixing".

Would be better placed in the results discussion. Page 13 line 24. "Sofiev et al (2013) also show observed birch pollen in Europe being negatively correlated with RH, and has very little correlation with observed wind speed, due to the competing effects of strength versus increased ventilation and mixing."

*13. Would it be helpful to include a statement concerning how much variation the smoothed statistical approach potentially might miss over and above the seasonal maxima?*

Move text from page 14 lines 4-7 to bottom of page 13 (keeps E10 discussion together). Include the following text: "Both the statistical emission parameterisations assume an underlying Cauchy distribution, which is modulated by the effects of wind, temperature, humidity and rainfall. At each model grid-cell, the peak and magnitude of this bell curve is calculated from statistics inferred from the EVI gradient."

Rewrite text on page 16 line 9. "The smoothed statistics for E10 used 16 years of observational data from the UoM and one year from the seven other Victorian sites. The smoothed statistical approach is modulated by the hourly effects of wind, temperature, RH and rainfall, which introduces temporal variation. The EVI also varies spatially and temporally, meaning that this method is suitable for future years and for other regions of Australia."

**References**

Asthma and Allergy Foundation of America. Asthma Capitals Risk Factors 2018. https://www.aafa.org/media/2119/aafa-2018-asthma-capitals-report.pdf

Beggs, P. J., Katelaris, C. H., Medek, D., Johnston, F. H., Burton, P. K., Campbell, B., Jaggard, A. K., Vicendese, D., Bowman, D. M. J. S., Godwin, I., Huete, A. R., Erbas, B., Green, B. J., Newnham, R. M., Newbigin, E., Haberle, S. G., and Davies, J. M.: Differences in grass pollen allergen exposure across Australia, Aust. NZ J. Publ. Heal., 39, 51–55, https://doi.org/10.1111/1753-6405.12325, 2015.

Coutts, A. M., Beringer, J., & Tapper, N. J. (2007). Characteristics influencing the variability of urban $CO_2$ fluxes in Melbourne, Australia. *Atmospheric Environment*, *41*(1), 51 - 62.

Damialis, et al. (2017), Estimating the abundance of airborne pollen and fungal spores at variable elevations using an aircraft: how high can they fly? Nature Scientific Reports, 7, Article number: 44535.

Erbas, B., Chang, J-H., Newbigin, E., Dhamarge, S. (2007) Modelling atmospheric concentrations of grass pollen using meteorological variables in Melbourne, Australia, International Journal of Environmental Health Research, 17:5, 361-368, DOI: 10.1080/09603120701628693

Haberle, S., Bowman, D., Newnham, R., Johnston, F., Beggs, P., Buters, J., Campbell, B., Erbas, B., Godwin, I., Green, B., Huete, A., Jaggard, A., Medek, D., Murray, F., Newbigin, E., Thibaudon, M., Vicendese, D., Williamson, G., and Davies, J.: The macroecology of airborne pollen in Australian and New Zealand urban areas, PLoS ONE, 9, e97 925, https://doi.org/10.1371/journal.pone.0097925, 2014.

Lake, IR, Jones, NR, Agnew, M, et al. Climate change and future pollen allergy in Europe. Environ Health Perspect. 2017; 125: 385- 391.

Osborne, N.J., Alcock, I., Wheeler, B.W. et al. Pollen exposure and hospitalization due to asthma exacerbations: daily time series in a European city. Int J Biometeorol (2017) 61: 1837. https://doi.org/10.1007/s00484-017-1369-2

Pasken, R. and J.A. Pietrowiez (2005) Using dispersion and mesoscale meteorological models to forecast pollen concentrations, Atmos. Environ., 39, 2689-7701.

Schäppi, G. F., Taylor, P. E., Kenrick, J., Staff, I. A., and Suphioglu, C.: Predicting the grass pollen count from meteorological data with regard to estimating the severity of hayfever symptoms in Melbourne (Australia), Aerobiologia, 14, 14–29, https://doi.org/10.1007/BF02694592, 1998.

Sofiev, M., Siljamo, P., Ranta, H., Linkosalo, T., Jaeger, S., Rasmussen, A., Rantio-Lehtimäki, A., Severova, E., and Kukkonen, J.: A nu35 merical model of birch pollen emission and dispersion in the atmosphere. Description of the emission module, International Journal of Biometeorology, 57, 45–58, https://doi.org/10.1007/s00484-012-0532-z, 2013.

US National Allergy Bureau. https://www.aaaai.org/global/nab-pollen-counts/reading-the-charts. American Academy of Allergy, Asthma and Immunology.

Van Hout et al., (2008) The influence of local meteorological conditions on the circadian rhythm of corn (Zea mays L.) pollen emission, agricultural and forest meteorology, 148, 1078-1092.

Viner, B.J., M.E. Westgate and R.W. Arritt, 2010: A model to predict diurnal pollen shed in maize. Crop Science 50, 235-245.

Zink, K., Pauling, A., Rotach, M. W., Vogel, H., Kaufmann, P., and Clot, B.: EMPOL 1.0: A new parameterization of pollen emission in numerical weather prediction models, Geosci Model Dev, 6, 1961–1975, https://doi.org/10.5194/gmd-6-1961-2013, 2013.

---

## Author Response (AR2)

Thanks Dave,

I have done a full read through of the paper, and think it is ok. I received instruction on how to include the bibliography within the .tex document, so the references should all be present.

On my read through I noticed I had not uploaded the new figure 4 as requested by reviewer #1 (only changes were the x axes numbering). I have corrected that now.

Thanks

Kathryn